# The ATR inhibitor ceralasertib potentiates cancer checkpoint immunotherapy by regulating the tumor microenvironment

The Ataxia telangiectasia and Rad3-related (ATR) inhibitor ceralasertib in combination with the PD-L1 antibody durvalumab demonstrated encouraging clinical benefit in melanoma and lung cancer patients who progressed on immunotherapy. Here we show that modelling of intermittent ceralasertib treatment in mouse tumor models reveals CD8[+] T-cell dependent antitumor activity, which is separate from the effects on tumor cells. Ceralasertib suppresses proliferating CD8[+] T-cells on treatment which is rapidly reversed off-treatment. Ceralasertib causes up-regulation of type I interferon (IFNI) pathway in cancer patients and in tumor-bearing mice. IFNI is experimentally found to be a major mediator of antitumor activity of ceralasertib in combination with PD-L1 antibody. Improvement of T-cell function after ceralasertib treatment is linked to changes in myeloid cells in the tumor microenvironment. IFNI also promotes anti-proliferative effects of ceralasertib on tumor cells. Here, we report that broad immunomodulatory changes following intermittent ATR inhibition underpins the clinical therapeutic benefit and indicates its wider impact on antitumor immunity.

The DNA damage response (DDR) is a basic mechanism by which a cell protects itself from deleterious consequences of DNA damage. DDR is regulated by a network of pathways with ATR being the apical signaling kinase in the DNA damage replication stress response (RSR) pathway[1]. ATR is recruited and activated by damaged and stalled replication forks. Recruitment and activation of ATR leads to replication fork stabilization, prevention of new origin firing and G2/M cell cycle arrest allowing time for DNA repair. Chronic inhibition of ATR in situations of high replication stress leads to replication fork collapse, accumulation of DNA damage, DNA fragmentation, chromosome mis-segregation, nuclear envelope breakdown, micronuclei formation and cell death. Recently, ATR and the RSR have been associated with activation of the innate immune response, leading to the priming of the adaptive immune system which may drive enhanced antitumor responses. These studies are primarily focused on tumor cell intrinsic extra-nuclear nucleic acid sensing mechanisms activated in response to tumor cell damage caused by ionizing radiation (IR)[2–5] or genomic instability in tumor cells[6–9]. Unrepaired DNA damage contributes to the formation of endogenous cytosolic DNA or RNA leading to the activation of cGAS-STING and/or RIG-I-MAVS pathways to induce IFNI signaling. However, the mechanism of the effects of ATR inhibition on the tumor immune microenvironment (TIME) and on immune cells themselves remained largely unclear.

Ceralasertib is a selective, oral inhibitor of the ATR kinase[10,11] currently in late-stage clinical development. Evidence of clinical activity for the combination of ceralasertib and anti-PD-L1 therapy (durvalumab) has been observed in multiple Phase II studies and across indications, including non-small cell lung cancer (NSCLC) patients who had progressed on prior anti-PD-(L)1-containing immunotherapy in the HUDSON trial[12,13], patients with advanced melanoma[14], and advanced gastric cancer[15]. However, the underlying mechanisms of this phenomenon remain unclear.

Here we show that ceralasertib in combination with anti-PD-L1 demonstrates strong antitumor effects in immunocompetent mouse tumor models, even in the absence of exogenous DNA damage (e.g., IR) or sensitizing tumor molecular alterations (e.g., ATM-loss).

e-mail: dmitry.gabrilovich@astrazeneca.com; simon.t.barry@astrazeneca.com

This activity is dependent on IFNI signaling and requires intermittent ATR inhibitor (ATRi) dosing to maintain the TIME in an immunotherapy responsive state.

## Results

### ATR inhibition resulted in CD8+T-cell dependent antitumor activity in mouse syngeneic tumor models

In clinical trials, ceralasertib is administered using an intermittent (7–14 day-on/14–21 day-off every 28 days) dosing strategy that is scheduled with administration of durvalumab (PD-L1 antibody). The impact of intermittent ceralasertib treatment in combination with anti-PD-L1 was studied in several subcutaneous (s.c.) and orthotopic syngeneic tumor models: CT26, MC38, 4T1, and A20. Consistent with the mode of action of an ATRi[11] 7-day treatment with a clinically relevant dose of 25 mg/kg b.i.d.(twice a day) ceralasertib induced the phosphorylation of the DNA damage response (DDR) markers KAP1 serine 824 (pKAP1) and H2AX serine 139 (γH2AX) in CT26 tumors, which returned to control levels upon cessation of drug treatment (Supplementary Fig. 1). We then modeled the clinical dosing regimen and used an intermittent 7 days-on/7 days-off dosing schedule. Under these conditions, monotherapy ceralasertib treatment reduced growth rate in all tumor models tested (Fig. 1a and Supplementary Fig. 2). Combination with PD-L1 antibody treatment demonstrated strong antitumor activity in all models (Fig. 1a). Because ceralasertib had potent antitumor activity as single agent, further inhibition of tumor growth with combination was observed in some (MC38, A20, CT26) but not others (4T1) tumor models. (Fig.1a and Supplementary Fig. 2). Combination therapy demonstrated more potent effect on the survival of CT26 tumor-bearing (TB) mice. While a small number of animals had longer survival with monotherapy anti-PD-L1 or ceralasertib treatment, the combination demonstrated substantially longer survival (Fig. 1b). In all models tested, depletion of CD8+ T-cells abrogated the antitumor effect of monotherapy ceralasertib treatment (Fig. 1c and Supplementary Fig. 3A, B) indicating that in the context of these models the antitumor effect of ceralasertib was CD8+ T-cell dependent. In nude mice lacking T-cells no antitumor efficacy was observed in MC38 or CT26 tumor-bearing mice (Fig. 1d and Supplementary Fig. 3C, D), confirming the contribution of T-cells to the antitumor effect of ceralasertib. CD8+ T-cells were required for sustained antitumor activity of ceralasertib as the depletion of CD8+ T-cells in the off-treatment period, after the first cycle of treatment, also reversed therapeutic benefit (Supplementary Fig. 3E).

Since ceralasertib activity was dependent on the presence of CD8+ T- cells and ATR inhibition is known to affect highly replicating cells, we evaluated changes in T-cells numbers during the on- and off-treatment phases. The treatment of mice with the intermittent 25 mg/kg b.i.d schedule of ceralasertib transiently decreased intratumoral CD8+ T-cells which rebounded after stopping ceralasertib treatment. In contrast, continuous treatment showed sustained reduction of intratumoral CD8+ T-cells as assessed by immunohistochemistry (IHC) staining (Fig. 1e). We compared the antitumor effect of continuous and intermittent dosing of ceralasertib in CT26 TB mice. Tumor response was classified using IN vivo reSPonsE Classification of Tumors (INSPECT). Intermittent dosing of mice with ceralasertib provided significantly better antitumor efficacy than continuous dosing (Fig. 1f, Supplementary Fig. 4A–C). This suggests that consistent with the importance of T-cells in mediating responses to ceralasertib, T-cell recovery and function during the off-treatment phase is essential to maximize therapeutic benefit.

In the MC38 TB mice, reduction of CD8+ T- cells in tumors and spleens in the on-treatment phase, was followed by the recovery during the off-treatment phase (Fig. 2a). Similar effects of ceralasertib regimens were seen for CD4+ T-cells (Supplementary Fig. 4D, E).

Consistent with these findings, similar reduction and recovery dynamics of T-cells were observed in advanced NSCLC patients treated with ceralasertib plus durvalumab in HUDSON trial[13,16,17]. To further characterize the mechanisms of action of ATR inhibition on T-cells, in vitro analysis was performed on CD3/CD28 stimulated, proliferating T-cells isolated from the peripheral bloods of healthy individuals. Three populations: CD4+ T-cells, memory and naive CD8+ T-cells were isolated and their response to ceralasertib examined. Ceralasertib reduced the accumulation of T-cells that had undergone more than 3 cell divisions (Fig. 2b, Supplementary Fig. 5A). This effect was dose dependent and ceralasertb $IC_{50}$'s were similar across naïve CD8+, memory CD8+ or CD4+ T-cell subtypes which demonstrates impact on all subtypes stimulated to proliferate (Fig. 2c). The differential sensitivity of proliferating T-cells to ceralasertib was likely to be an on-target ATR related mechanism as western blot protein expression analysis of T-cells showed that ATR expression was undetectable in unstimulated (non-replicating) T-cells, consistent with the lack of ATR inhibitor activity on this population (Fig. 2d). Upon stimulation, ATR protein expression was induced along with phosphorylation of RPA (pRPA) pS4/8 and γH2AX which are markers of replication-associated DNA breaks (Fig. 2d). This upregulation of ATR potentially renders T-cells more dependent on ATR to initiate repair of damaged DNA, which accumulates in cells that have undergone more cell divisions (replications)[18,19]. Consistent with the known consequences of ATR inhibition, ceralasertib treatment of stimulated T-cells for 4 days further increased DNA damage as evidenced by strong induction of pRPA, pKAP1, pCHK2 and p53 protein DDR markers to similar levels as positive control etoposide DNA-break inducing chemotherapy treatment (Fig. 2d, Supplementary Fig. 5C). The specific loss of highly replicating T-cells observed in humans and mice was consistent with ceralasertib targeting T-cells with high replication stress (Supplementary Fig. 5C). The anti-proliferative effect was reversible, and T-cells cells re-grew following removal of ceralasertib (Fig. 2e). Collectively, these data demonstrated that the antitumor effect of ceralasertib in mouse models was mediated by T-cells and thus required an intermittent schedule to allow for a quick recovery of T-cell proliferation.

### ATR inhibition reshaped the tumor immune microenvironment

We further examined the T-cell populations associated with antitumor efficacy of ceralasertib. Different populations of T-cells, identified by cell surface protein marker expression, were analyzed by unbiased clustering analysis using single cell mass cytometry (CyTOF) in tumors from CT26 TB mice (a T-cell rich tumor model) treated with ceralasertib alone or in combination with anti-PD-L1 after one 7-days on/7-days off cycle (Fig. 3a). Following ceralasertib monotherapy treatment a significant increase in the presence of resting CD8+ T-cells (PD1−, GzmB−) was observed, while the population of CD8+ T-cells with markers characteristic of exhaustion (PD1+, 41BB+, ICOS+, Lag3+, GzmB^low, IFNγ−) was decreased relative to control and PD-L1 antibody treated mice. The magnitude of these changes were further potentiated when ceralasertib was combined with anti-PD-L1. The early activated effector CD8+ T-cells (PD1+, GzmB^high) were minimally affected (Fig. 3b). T-cells isolated from tumors of non-treated and ceralasertib only treated mice (at the end of 7 days-on/7 days-off cycle) were also examined using single-cell transcriptomics analysis (scRNAseq) in an independent study (Fig. 3c). Similar to the findings with CyTOF analysis, a trend for a reduction in the cell clusters expressing signatures of activation or exhaustion was observed, although this did not reach statistical significance, and a reciprocal statistically significant increase in resting or naïve cells (CCR7+Sell+) (Fig. 3d).

In the MC38 TB mice at the end of the off ceralasertib monotherapy treatment period (7 days-on/7days-off), ceralasertib treatment slightly reduced CD8+ T-cells in the tumor and spleen (Supplementary Fig. 6A) and showed minimal effects on the effector or memory CD4+ or CD8+ T-cells similar to the CT26 model, although T-cell numbers were low in MC38 tumors (Supplementary Fig. 6B, C). In spleens, the

 

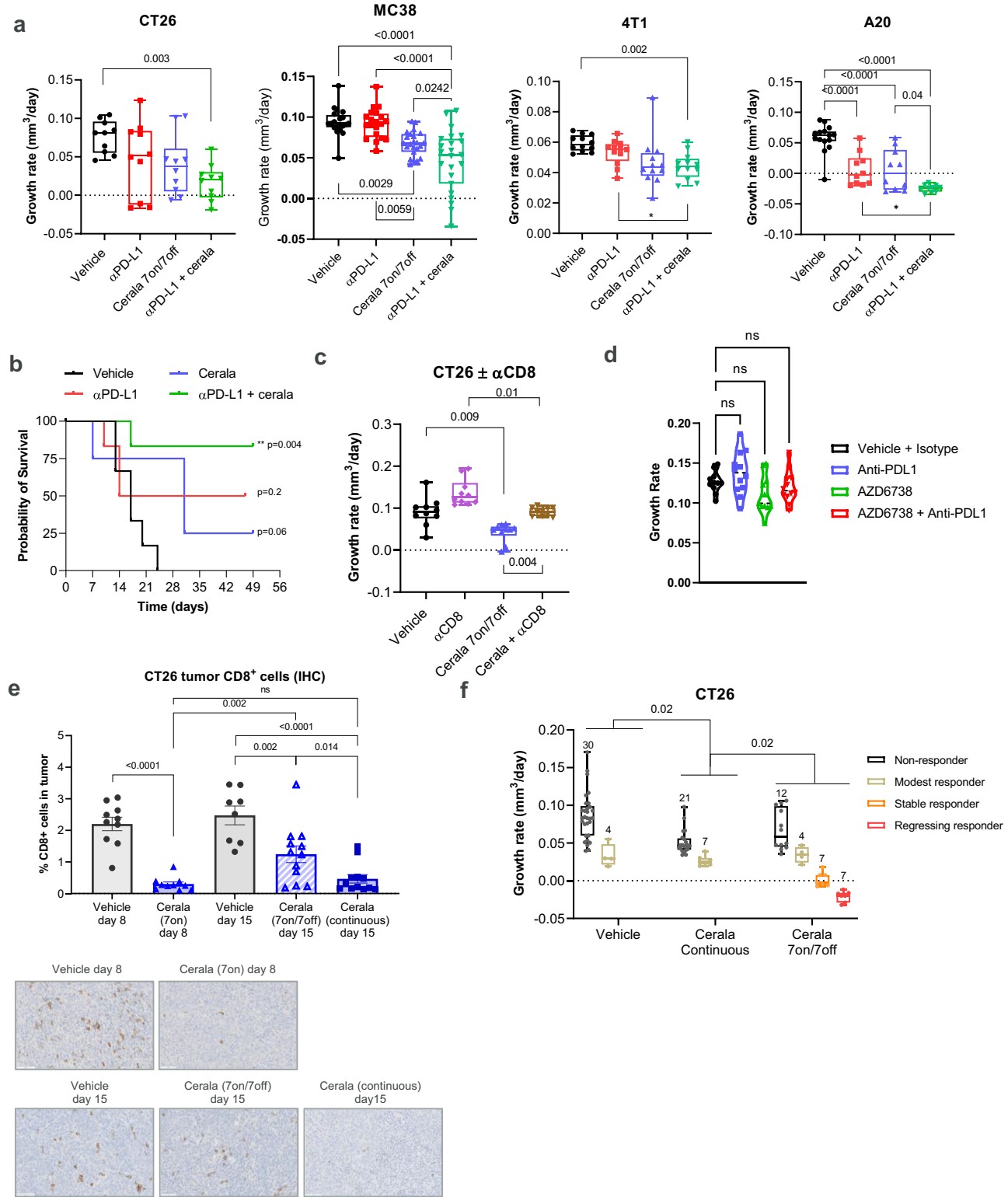

monotherapy ceralasertib treatment increased CD44^lo/inter CD62L⁻ effector CD8⁺ T-cells (Supplementary Fig. 6C).

Next, we assessed changes in thymic precursors during the treatment. Seven-day treatment of MC38 TB mice with ceralasertib caused a marked decrease in the presence of all populations of T-cell precursors in the thymus (Supplementary Fig. 7A), as well as all populations of CD4/CD8 double negative (DN) cells (Supplementary Fig. 7B). Seven days off ceralasertib treatment recovery in the populations of thymocytes was evident, for both later stages of maturation

of DN (DN-2-4, Supplementary Fig. 7B) and double positive thymocyte (Supplementary Fig. 7A) populations. Single positive CD4 thymocytes also demonstrated significant recovery (Supplementary Fig. 7A). These results were consistent with T-cell recovery in the periphery after the treatment.

Although the presence of PD1⁺ CD8⁺ T-cells was not changed, ceralasertib treatment induced a substantial increase in the expression of PD1 in the total population of tumor CD4⁺ T-cells as well as CD4⁺ CD44^lo/- CD62L⁺ T effector cells. There was a trend in increased PD1

**Fig. 1 | ATR inhibition has immune driven CD8+ T-cell dependent antitumor activity in syngeneic immunocompetent mouse tumor models. a** Tumor growth rate (TGR) for CT26 (N = 10), MC38 (N = 20 for vehicle and PD-1 groups and 23 for two other groups), 4T1 (N = 12), and A20 (N = 15 for vehicle group and N = 10 for all other groups). TB mice treated as indicated. Ceralasertib (25 mg/kg p.o. b.i.d.), PD-L1 antibody (αPD-L1; 10 mg/kg i.p. b.i.w) **b** Kaplan−Meier survival (time-to-event) plot for mice bearing CT26 tumors. An event was scored when a tumor volume exceeded 1 cm³. Log-rank (Mantel−Cox) test was used. **c** TGR for CT26 tumors in mice treated with ceralasertib with or without CD8 T-cell depleting antibody (αCD8). (N = 10). **d** Tumor growth in athymic MC38 TB mice treated as in (**a**). (N = 10). **e** Immunohistochemistry (IHC) analysis of proportion of CD8+ T-cells in CT26 tumors from mice treated with vehicle (N = 9) or ceralasertib (N = 10) following 7 days-on (on day 8), intermittent 7 days-on/7 days-off (on day 15) (vehicle

N = 8, cerala N = 12) or continuously on-treatment for 15 days (N = 12). Top panel: cumulative results with bar charts showing individual tumors, mean and SEM. Bottom panel: representative images also shown with CD8+ T-cells stained brown. Scale bar = 50 μm. **f** Combined tumor growth rate analysis in 3 independent experiments comparing intermittent 7 days-on/7 days-off versus continuous ceralasertib dosing in CT26 TB mice. The number of mice per group are shown on the plot. Tumor response classification was obtained using INSPECTumors (see methods). Non-significant differences (p > 0.05) are denoted as NS or by the absence of p values (to maintain readability of the figures). In all graphs except b, one-way ANOVA with correction for multiple comparisons was used for statistical analysis. In all graphs box-whisker plots shows individual tumors, group median and min-max values. Source data are provided as a Source Data file.

expression in tumor CD8+ cells which did not reach statistical significance. (Supplementary Fig. 7C). In spleens, expression of PD1 was markedly up-regulated in CD8+ T-effector cells (Supplementary Fig. 7C).

Further exploration of the impact of ceralasertib treatment on tumor myeloid cell subtypes revealed that after 7 days of ceralasertib monotherapy minimal changes in immune suppressive (Supplementary Fig. 8A, B) polymorphonuclear-myeloid derived suppressor cells (PMN-MDSC) were observed. However, ceralasertib treatment caused a significant decrease in the presence of immune suppressive monocytic-MDSC (M-MDSC) and tumor associated macrophages (TAM) (Supplementary Fig. 8A, B). Similar results were observed in spleens (Supplementary Fig. 8C). Ceralasertib treatment caused increase in the percentage of tumor CD11c+MHC II+ dendritic cells (DCs) and marked up-regulation of the expression of molecules associated with DC activation: MHC class II (IAb), MHC class I (H2Kb), CD40, and CD80 (Supplementary Fig. 8D). In addition, analysis of myeloid subtypes in CT26 tumors from ceralasertib treated mice also showed a significant decrease in the number of tumor associated macrophages and increased percentage of CD11c+MHC II+ DCs (Supplementary Fig. 8E). Thus, ceralasertib treatment depleted some immune suppressive populations of myeloid cells and showed evidence of activated DCs and increased expression of PD1 on T cells potentially making them more susceptible to check-point inhibitors. This effect was not confined just to TIME but was observed systemically.

The effect of ceralasertib on myeloid cell progenitors in bone marrow was also assessed. Treatment with ceralasertib did not significantly affect the presence of myeloid progenitors (Supplementary Fig. 9A, B), which was consistent with lack of the effect on the presence of largest population of myeloid cells in periphery (PMN).

To determine whether differential effects of ceralasertib on T cells and myeloid cells reflected differences in ATR expression we isolated cells from spleens of TB mice and assessed different components of the DDR machinery. ATR and ATM as well as other components of the DDR machinery or cycle cycle-proliferation were highly expressed in T cells. In monocytes and macrophages total ATR, ATM, CHK1 and H2AX protein as well as CyclinE were clearly detectable (albeit at lesser degree than in T cells). However, PMN were negative for ATR expression and a number of other DDR and cell cycle-proliferation biomarkers (Supplementary Fig. 9C). Human PMN cells also did not express ATR or cell cycle markers. However, taking isolated human monocytes into culture, and then stimulating differentiation with different cytokines resulted in upregulation of ATR (Supplementary Fig. 9D). Collectively, these data reinforce the notion that actively replicating T-cell or some myeloid cell populations upregulate the ATR dependent replication stress pathway, required for unperturbed cell proliferation and genome stability, and thus may be selectively susceptible to ATR inhibition. The lack of ATR expression in PMN may explain the lack of the effect of ceralasertib on PMN numbers in vivo.

## The effect of ATR inhibition on immune microenvironment can be independent on its effect on tumor cells

To better understand the mechanism of ceralasertib, we investigated whether ceralasertib mediated reshaping of the TIME was dependent on its direct effect on tumor cells. To address this question, we used tumor-free Lymphocytic Choriomeningitis Virus (LCMV) (clone 13) model. Ceralasertib treatment was started 11 days after the virus injection and continued for 7 days (Supplementary Fig. 10A). Gp33 tetramers were used to detect LCMV specific CD8+ T-cells. Ceralasertib alone or PD-L1 antibody alone did not cause expansion of IFN-γ producing antigen-specific CD8+ T-cells. However, the combination resulted in increase in the presence of these cells in spleen (Fig. 4a) and blood (Supplementary Fig. 10B). Moreover, combination treatment, but not single treatments, induced accumulation of polyfunctional (IFN-γ+ TNFα+) tetramer-positive CD8+ T-cells (Fig. 4b). Thus, the combination of ceralasertib and PD-L1 antibody resulted in stimulation of antigen-specific immune responses in tumor-free mice. Ceralasertib induced substantial increase in the presence of PD1+ gp33 tetramer positive CD8+ cells (Fig. 4c). These results recapitulated data in TB mice and indicated that inhibition of ATR with ceralasertib induced PD1 expression on antigen-specific T-cells that would make them more sensitive to the effect of PD-L1 blockade.

In all tested tumor models, ceralasertib had antitumor effect as monotherapy (Fig. 1 and Supplementary Fig. 1). We asked if the antitumor effect of combination of ceralasertib with anti-PD-L1 was dependent on the antitumor activity of ceralasertib as monotherapy. To address this question, we determined the highest concentration of the drug that did not affect the tumor cell proliferation in vitro (Supplementary Fig. 11A). This dose was 4-fold lower than optimal antiproliferative concentration. Therefore, we tested in vivo 6.25 mg/kg dose, which was 4-fold lower than optimal dose (25 mg/kg). At this dose, ceralasertib caused weaker but still detectable DDR effect in vivo (Supplementary Fig. 11B, C) and had weak or undetectable antitumor activity as single agent (Supplementary Fig. 12A). However, combination with PD-L1 antibody still caused substantial antitumor activity, significantly stronger than the effect of ceralasertib alone (Supplementary Fig. 12A–D). Depletion of CD8+ T-cells abrogated antitumor effect of combination therapy (Supplementary Fig. 12C). Thus, ceralasertib effect on the TIME may not be dependent on its direct effect on tumor cells.

## Ceralasertib induced Type I IFN signaling pathways, which contributed to antitumor activity

We next explored the specific mechanisms of ceralasertib effect by performing an unbiased bulk RNAseq transcriptomic analysis of CT26 tumors from TB mice treated with ceralasertib, anti-PD-L1 or their combination, and when on-treatment (7 days-on) or in the off period at the end of the cycle (7 days-on/7days-off). It revealed substantial differences in gene expression profiles, with the most prominent amongst those up-regulated after ceralasertib treatment was the type I

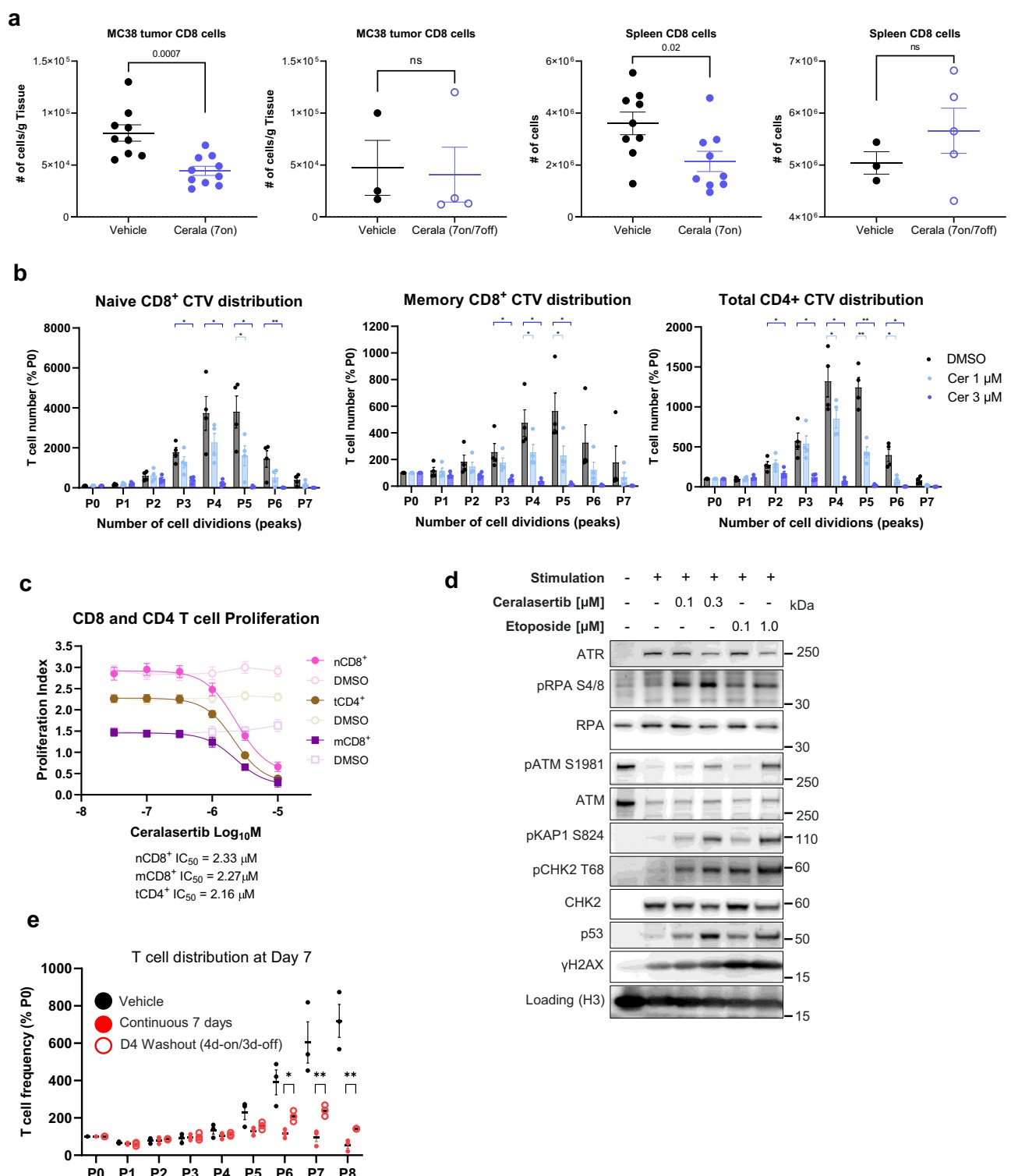

interferon, which remained strongly up-regulated in mice treated with combination of anti-PD-L1 and ceralasertib (Supplementary Fig. 13). More detailed analysis of IFNI pathway demonstrated that up-regulation was observed primarily after completion of 7-days on/7-days off intermittent dosing cycle (Fig. 4d, e).

In NSCLC patients treated with ceralasertib plus durvalumab in the HUDSON trial peripheral blood RNAseq gene expression analysis showed significant up-regulation of IFN signatures and expanding T-cell clones[17].

Activation of IFNI pathway by ATRi is not surprising considering possible role of ATRi in STING activation[4,6,20]. Down-regulation of STING abrogated ceralasertib induced IFNI production by tumor cells (Supplementary Fig. 14A, B) supporting this possibility. To determine the causal role of IFNI in regulation of antitumor effect of ceralasertib, we performed several experiments. First, the animals bearing CT26 tumors were treated with monotherapy ceralasertib at 25 mg/kg b.i.d, with or without an IFNAR1 blocking antibody. This treatment atte-nuated ceralasertib antitumor efficacy (Fig. 5a, Supplementary

**Fig. 2 | ATR inhibition selectively suppresses proliferating T-cells in humans and mouse. a** The number of CD3[+] CD8[+] T-cells in MC38 tumors and spleens following treatment with vehicle or ceralasertib for 7 days (N = 9 for tumor and spleen) or 7 days-on/7 days-off (on day 15), (N = 4 for tumor, N = 5 for spleen). The number of cells for each subset was calculated per gram of tissue. Individual values, mean and SEM are shown. *P* values are shown on graph. NS not significant (*p* > 0.05). **b** Anti-proliferative activity of ceralasertib at indicated concentrations against in vitro stimulated/proliferating (CD3/CD28) T-cells from PBMCs isolated from different healthy donors (N = 4). Numbers represent proportion of T-cells in each cell division (P1 = 1 cell division, P2 = 2 cell divisions etc.) compared to non-proliferative cells (P0) after 6 days treatment with ceralasertib using CellTrace Violet (CTV). One way ANOVA test with correction for multiple comparisons between treated and DMSO vehicle control samples. Experiments were repeated once with 4 additional donors with the same results. **c** Anti-proliferative activity of ceralasertib vs vehicle (DMSO) in naïve CD8[+] T-cells (nCD8[+]), memory CD8[+] T-cells

(mCD8[+]) and CD4[+] T-cells (tCD4[+]). Four individual donors were tested. Experiment was repeated once with 4 additional donors with the same results. Individual results, mean and SEM are shown. **d** Western blot protein expression of DNA damage and cell cycle markers in unstimulated (naïve/resting) or in vitro stimulated/proliferating T-cells following 4 days vehicle (DMSO), ceralasertib, or etoposide treatments at the concentrations indicated. Representative experiment from one donor is shown. Four unique donors were profiled in parallel in one experiment showing the same results. Experiments were repeated once with the same results. **e** CTV proliferation assay for stimulated/proliferating T-cells after 7 days continuous or intermittent 4 days-on/3-days off ceralasertib exposure. Cells from 3 individual donors were tested. Individual results, mean and SEM are shown. Two-sided unpaired Student's *t*-tests are shown as indicated. In (**b, e**) *p* values are shown as asterisk to maintain readability *\*p* < 0.05, \*\**p* < 0.01, \*\*\**p* < 0.001. Non-significant *p* values are not shown. Source data are provided as a Source Data file.

Fig. 14C). Second, the efficacy of ceralasertib and the combination with PD-L1 antibody was compared in wild-type (WT) mice and mice lacking the IFNAR1 chain of the IFNI receptor. We generated bone marrow (BM) chimeras by transferring BM from wild-type (WT) or IFNAR1 knockout (KO) mice to congenic lethally irradiated WT recipients (Supplementary Fig. 15A). Given the strong monotherapy activity of ceralasertib at 25 mg/kg b.i.d. in the MC38 model, to enable assessment of the benefit with anti-PD-L1 to be visualized ceralasertib was dosed at 6.25 mg/kg b.i.d. Eight weeks after transfer, the chimerism (>90%) was confirmed and MC38 tumor cells were injected s.c. Similar to the results in WT mice (Supplementary Fig. 2), in mice reconstituted with WT BM, treatment with anti-PD-L1 alone or with ceralasertib alone did not have potent antitumor activity. However, significant (*p* < 0.05) antitumor effect was observed in combination treatment group (Fig. 5b). This effect was notably absent in mice reconstituted with BM from IFNAR1 KO mice (Fig. 5b and Supplementary Fig. 15B). To determine if the observed up-regulation of PD-1 expression in T cells after ceralasertib treatment (Supplementary Fig. 7C) could be mediated by IFNI. CD4 and CD8 T cells isolated from spleens of naïve mice were treated with IFN-β in the presence of CD3/CD28 activation. IFN-β induced marked up-regulation of PD1 expression on T cells (Fig. 5c). MC38 TB wild-type and IFNAR1 knockout mice were treated with ceralasertib for 7 days and the expression of PD1 was assessed in T cells from spleens and tumors. In contrast to wild-type mice (Supplementary Fig. 7C), ceralasertib treatment did not up-regulate PD1 expression in IFNAR deficient T cells (Fig. 5d). These results strongly indicate that PD1 up-regulation in ceralasertib treated mice was mediated by IFNI.

Finally, the ability of a gain of function of IFNAR1 to enhance ceralasertib monotherapy treatment was explored. Efficacy was compared in mice with reconstituted BM from WT and IFNAR1[S526A] mutant knock-in mice (referred as SA mice) expressing stabilized IFNAR1 protein that is resistant to ubiquitination and degradation because it lacks critical Ser526 phosphorylation which enables the recruitment of β-TrCP E3 ligase[21,22]. Naïve *Ifnar1[SA]* mice maintained IFNAR1 expression even when challenged with inflammation[23], viral[24] or bacterial[25] infection and tumors[26] and thus are highly sensitive to IFNI. BM cells were transferred to lethally irradiated naïve recipient and 8 weeks after the transfer and confirmation of chimerism (>90%) mice were challenged with MC38 tumor cells and treated with ceralasertib at 6.25 mg/kg bid. As expected with 6.25 mg/kg bid ceralasertib treatment, no antitumor effect was observed in mice reconstituted with WT BM. In contrast, in mice reconstituted with SA BM, ceralasertib delivered strong antitumor activity (Fig. 5e). Taken together, these results indicate that ceralasertib potentiates antitumor effect of immune check-point blockade via up-regulation of IFNI signaling.

To determine whether ceralasertib is effective beyond transplantable syngeneic tumor models a GEMM model of non-small lung (NSCLC) cancer induced in OdIN mice was used. This model utilizes

doxocyclin inducible CAS9 mice[27] where lung tumors are induced by exposure of lungs to AAV containing RNA CRISPR guides to delete *p53* and mutate *Kras* (*Kras[G12C]*). Following viral induction mice developed lung tumor lesions within 6–7 weeks. Mice were treated with ceralasertib (7 days on/7 days off) starting 11.5 weeks post-tumor induction, following confirmation of lung tumor burden by MRI. Ceralasertib treatment significantly prolonged the survival of lung tumor-bearing mice (median survival: vehicle: 89 days; cerala: 101 days) (Fig. 6a). Following a 7-days on/7-days off treatment cycle, GEMM lung tumors were collected and their TIME profiled by flow cytometry. Treatment with ceralasertib caused significant decrease in alveolar macrophages and no changes proliferating T cells (Fig. 6b). Consistent with the observations in transplantable syngeneic models ceralasertib treatment induced up-regulation of IFNI signature genes in treated tumors (Fig. 6c).

## Mechanism of ceralasertib mediated IFNI regulation of TIME

In order to assess how can IFNI affects the TIME we evaluated tumor-specific function of T-cells. Splenocytes from MC38 TB mice (25 mg/kg b.i.d. ceralasertib 7 days-on/7 days-off) were collected and stimulated with either MC38 derived H2K[b] matched p15e peptide or irrelevant control gp100 peptide. T-cell response was measured in IFN-γ ELISPOT. Ceralasertib induced increase in tumor antigen-specific CD8[+] T-cells. This effect was completely abrogated in IFNAR1-KO mice (Fig. 7a). To assess the effect of ceralasertib on non-specific T-cell activation in the context of anti-PD-L1 treatment, CD8[+] T-cells isolated from mice following anti-PD-L1 or ceralasertib plus anti-PD-L1 combination treatment for a 7 days-on/7 days-off cycle were stimulated with PMA-Ionomycin. The proportion of IFN-γ producing CD8[+] T-cells was markedly greater in the combination treated animals following PMA-ionomycin stimulation (Fig. 7b). These data are consistent with ceralasertib treatment resulting in greater activation of T-cells present after the off-drug treatment phase, with ceralasertib monotherapy promoting activated IFN-γ producing antigen-specific CD8[+] T-cells, whereas in combination with the PD-L1 antibody it also induced activation of all CD8[+] T-cells.

In the MC38 model ceralasertib up-regulated the activation markers on tumor DCs (Supplementary Fig. 7). To determine functional activity of these cells, DCs were isolated from draining lymph nodes of MC38 tumor-bearing mice following 7 days of 25 mg/kg b.i.d. ceralasertib treatment and used for stimulation of allogenic CD4[+] or CD8[+] T-cells. We observed substantially higher stimulatory activity of DCs from ceralasertib treated mice (Fig. 7c). This effect was abrogated in IFNAR1 KO mice (Fig. 7d).

To directly assess the effect of ceralasertib on DC function, DCs were generated from BM of naïve or IFNAR1 KO mice using GM-CSF and IL-4 and then treated with different concentrations of ceralasertib for 48 h. Lipopolysaccharide (LPS) (10 ng/ml) was used as positive control. Ceralasertib caused marked increase in the

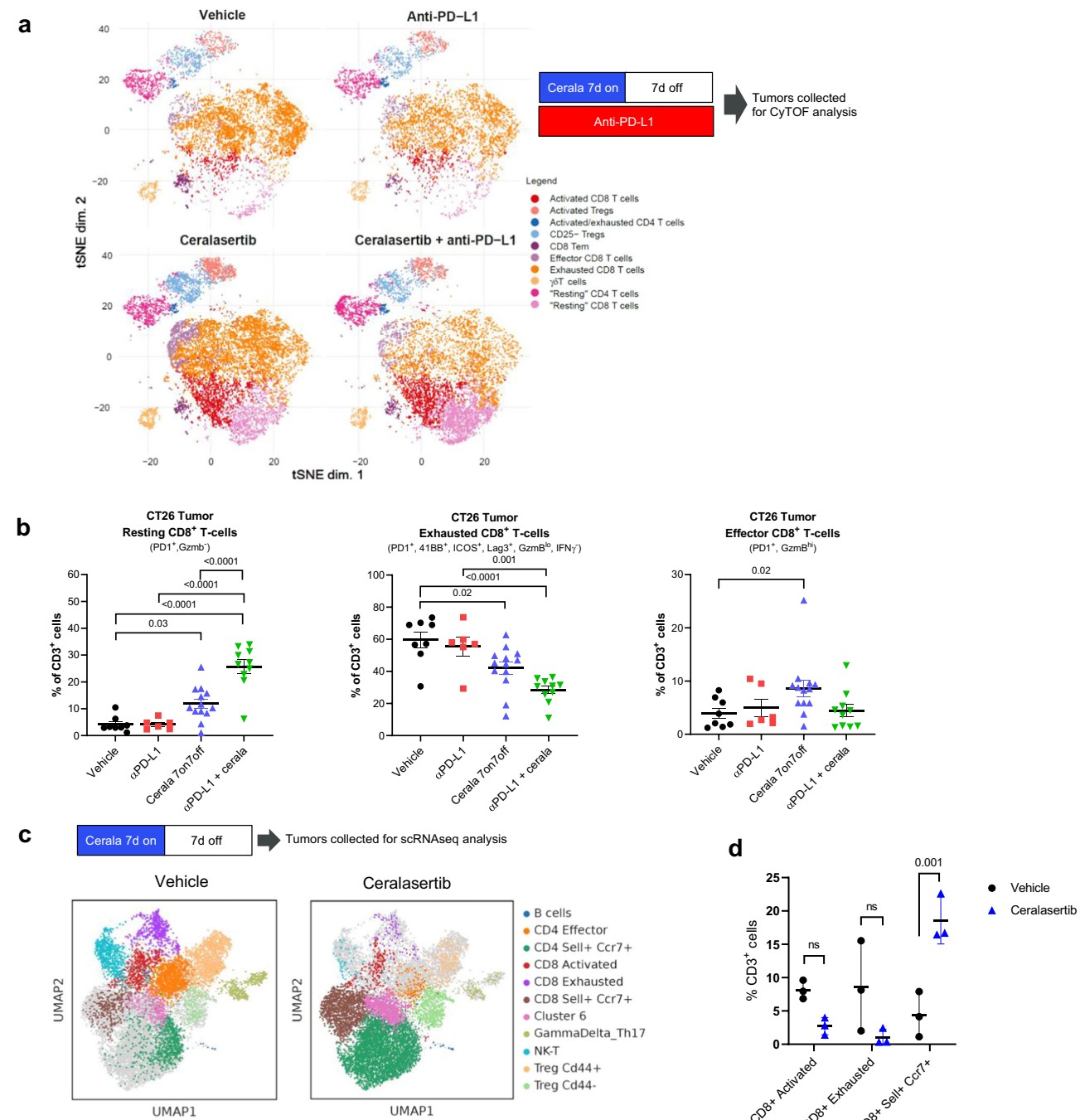

**Fig. 3 | Intermittent ATR inhibition reshapes the tumor immune micro-environment which promotes immunotherapy activity. a, b** CyTOF single cell immunophenotyping of CD45⁺CD3⁺ immune populations in subcutaneous CT26 tumors (*n* = 10 per group) of mice treated with 25 mg/kg b.i.d. ceralasertib, 10 mg/kg anti-PD-L1 b.i.w. or their combination at the end of a 7 days-on/7 days-off treatment cycle (day 14). **a** Study design and t-SNE plots visualization of high level unbiased clustering analysis of CD3⁺ T-cell populations classified by cell surface marker expression as indicated. **b** Quantification of the proportions of resting, exhausted or effector CD8⁺ T-cell populations as a percentage of total CD3⁺ cells which were significantly modulated following treatment compared to vehicle control. Individual tumor values, group mean and SEM are shown. Statistical

analysis performed by β-regression (R). **c, d** Single cell RNAseq analysis of CD3⁺ cells isolated from the tumors at the end of a 7 days-on/7 days off 25 mg/kg b.i.d. ceralasertib monotherapy treatment cycle (day 14) in CT26 tumors (*n* = 3 per group). **c** Study design and UMAP plot visualization of transcriptomic profiles of CD3⁺ expressing cells according to Leiden clustering. **d** Proportions of CD8⁺ T-cells populations as a percentage of the total CD3⁺ expressing cells per Leiden cluster vehicle and ceralasertib. Individual tumor values and group mean and SEMs are shown. Statistical analysis performed by one-way ANOVA with correction for multiple comparisons. In all panels ns means not significant (*p* > 0.05). *P* values are shown on the graphs. Source data are provided as a Source Data file.

expression of molecules associated with DC activation at concentrations as low as 250 nM and at higher concentration could reach the level comparable to that in the presence of LPS (Supplementary Fig. 16A). In the absence of IFNAR1 ceralasertib failed to activate DCs.

Only at higher concentrations of 2 µM, it caused up-regulation of CD86 but no other molecules (Supplementary Fig. 16B). DCs treated with ceralasertib induced potent activation of allogenic CD8⁺ T-cells similar to the effect of LPS (Supplementary Fig. 16C). Thus, treatment

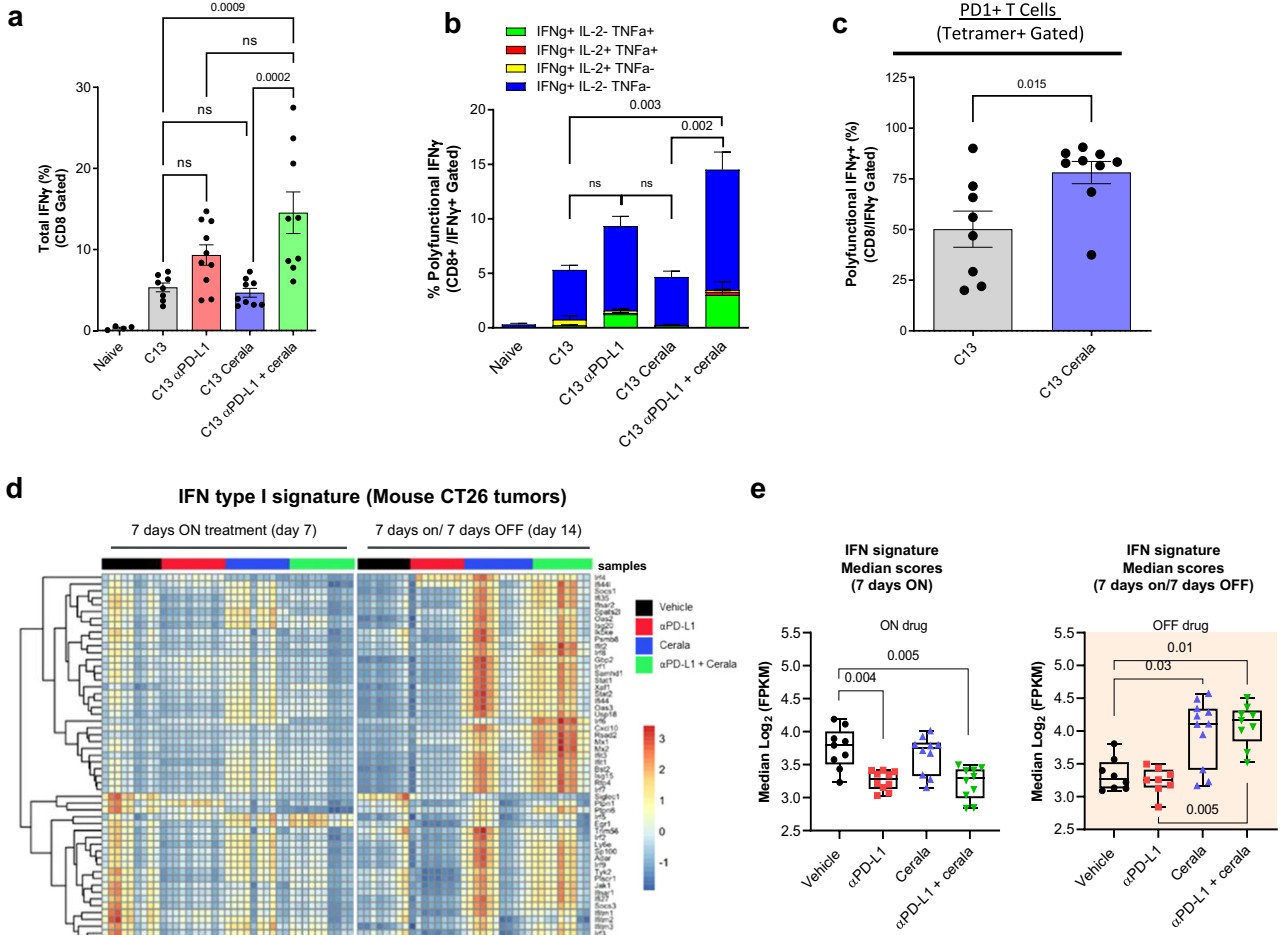

**Fig. 4 | Ceralasertib effect in LCMV infected and tumor-bearing mice.**
**a** Proportion of gp33⁺IFNγ⁺ CD8⁺ T-cells in the spleen of LCMV infected mice treated as indicated on graph n = 10. Individual results, mean and SEM are shown. P values are calculated in one-way ANOVA with correction for multiple comparisons and shown on graphs. NS−p > 0.05. **b** Percentages of gp33⁺ CD8⁺ T-cells producing IFNγ, TNFα, and IL-2 were evaluated by intracellular staining. Biological replicates (mice) N = 10 are shown. Mean and SEM are shown. P values are calculated in one-way ANOVA with correction for multiple comparisons and shown on graphs. Not significant (p > 0.05). **c** PD1 expression on gp33⁺ CD8⁺ T-cells. Individual results (mice, N = 8), mean, and SEM are shown. P values were calculated in two-sided unpaired Student's t-test. **d**, **e** Bulk RNAseq analysis of CT26 tumors from mice treated with 25 mg/kg b.i.d. ceralasertib, 10 mg/kg anti-PD-L1 b.i.w. or their

combination when on ceralasertib treatment (7 days-on; day 7) and when off-treatment (at the end of 7 days-on/7 days-off cycle; day 14) as indicated. N = 8 for C13, N = 9 for C13 cerala. **d** Heatmap of Interferon type I signature gene expression, calculated from FPKM normalized counts expressed above biological viable levels, log-transformed and normalized for gene lengths. The data was z-transformed to be visualized in heatmap. **e** Box plots of median scores and maximum/minimum of interferon type I signature after 7 days on-treatment, or 7 days on/7 days off-treatment (N = 8). The medians were calculated of the log2-transformed FPKM for each sample and significance calculated using Fisher exact T-test. In all panels ns means not significant (p > 0.05). P values are shown on graphs. Source data are provided as a Source Data file.

with ceralasertib caused IFNI dependent activation of T-cells and DCs.

Myeloid cells are major negative regulators of T-cell function in the TIME. Treatment with ceralasertib caused marked reduction of the numbers of immune suppressive M-MDSC and macrophages in tumors and spleen (Supplementary Fig. 7). This could reduce immune suppressive effect of TIME. However, no changes in the numbers of PMN-MDSC was observed (Supplementary Fig. 7). We asked if ceralasertib treatment could instead affect suppressive function of PMN-MDSC. Tumor PMN-MDSC were assessed in antigen-specific T-cell suppression assay. As expected, PMN-MDSC demonstrated potent suppressive activity of CD8⁺ T-cells. Treatment with ceralasertib markedly reduced this activity (Fig. 7e). This was associated with up-regulation of IFNI pathway in these cells (Fig. 7f).

Having established that IFNI is induced in the TIME, and mediates activation of different immune cells we assessed whether IFNI can impact the effect of ceralasertib on tumor cells. Treatment of CT26, MC38 mouse tumor cells with 1 ng/ml and A20 with 0.3 ng/ml IFN-β

resulted in reduction in tumor cell proliferation in vitro (Fig. 8a). As expected, there was generally modest anti-proliferative effects following ceralasertib treatment at 1 μM. However, when IFN-β and ceralasertib were combined together there was a marked reduction in tumor cell proliferation. The effect was reduced by the treatment with a JAK inhibitor (5 μM) which prevented signaling downstream from IFNAR1. Moreover, the combination activity was not observed in MC38 cells not expressing IFNAR1 (Fig. 8a). This suggested that in addition to promoting immune cell activation, IFNI induced by ceralasertib in the TIME can directly impact the proliferation of tumor cells. To determine whether this effect was applicable to human cells, a panel of human NSCLC cell lines were treated with ceralasertib, IFN-β, and their combination (Fig. 8b). Thirteen out of sixteen lines showed significantly enhanced growth inhibition with the combination when compared to vehicle (DMSO), eight out of sixteen when compared to ceralasertib monotherapy and three of sixteen cell lines when compared to IFN-β monotherapy. The combination activity was reversed by the treatment with a JAK inhibitor (Fig. 8b and statistics in Supplementary Fig. 17A).

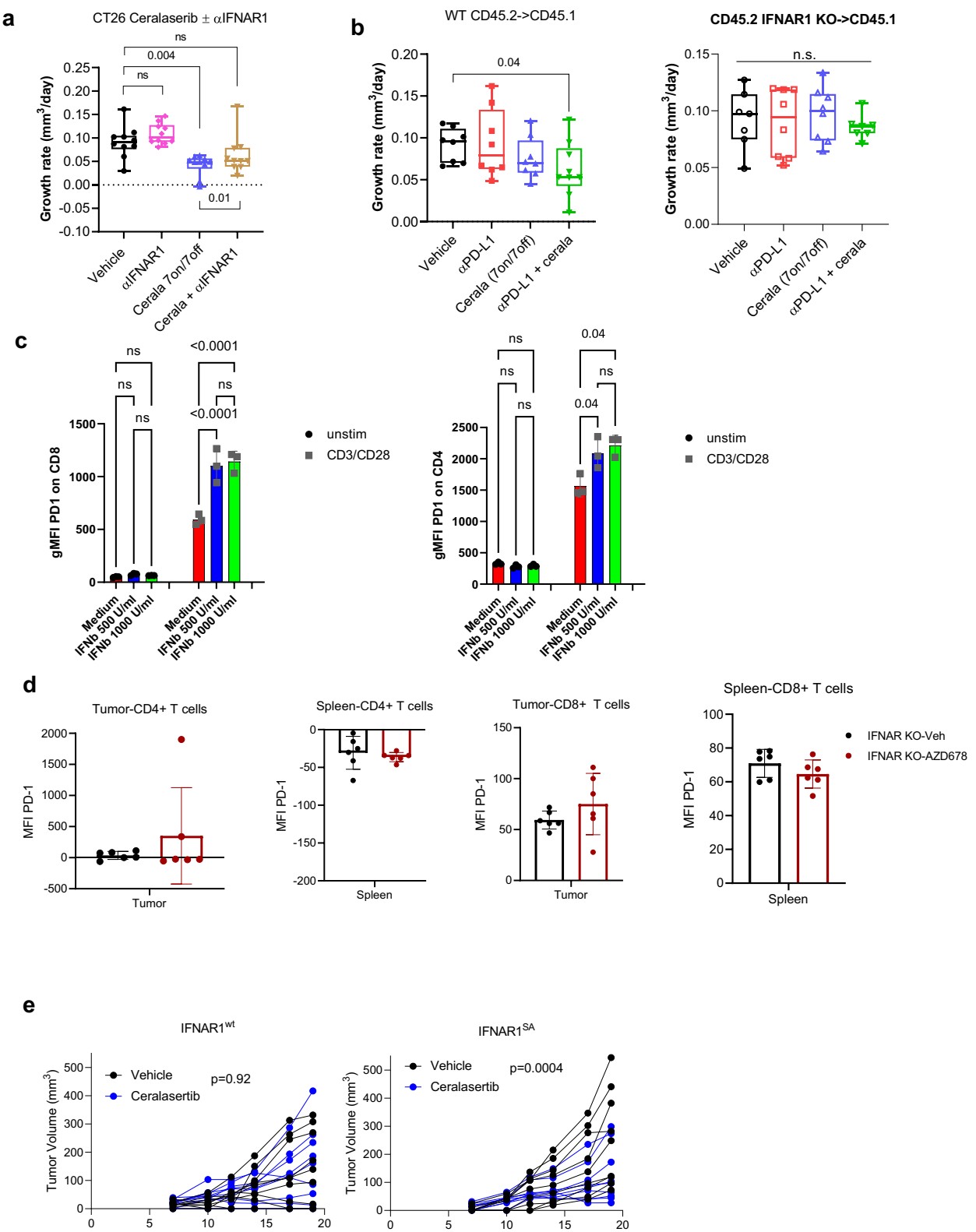

Furthermore, the anti-proliferative effect of ceralasertib in combination with IFN-β was abrogated by siRNA mediated knockdown of IFNAR1 expression in tumor cells (Fig. 8c, Supplementary Fig. 17B). The NCI-H23 NSCLC cell line was used as a control and was known to be highly sensitive to ceralasertib monotherapy which has been attributed to an ATM-loss DDR-defect[11,17] and growth inhibition could not be further increased when combined with IFN-β nor abrogated by IFNAR1 knockdown. Western blot protein analysis showed induction of target engagement markers pKAP1 and pSTAT1 following ceralasertib and IFN-β treatment respectively in mouse and selected human cell lines (Fig. 8d, e and Supplementary Fig. 17C). γH2AX was also induced following ceralasertib treatment in all cell lines analyzed. In addition, evidence of increased induction of apoptosis markers cleaved-PARP1, cleaved-caspase 8 and cleaved-caspase 3 following ceralasertib and

**Fig. 5 | IFNI mediates the effect of ceralasertib in tumor microenvironment.**
**a** TGR of CT26 tumors in mice treated with ceralasertib 7 days-on/7 days-off and with an IFNAR1 blocking antibody b.i.w. Values for individual mice ($N = 10$), median, and min-max values are shown. Statistical analysis was performed using one-way ANOVA corrected for multiple comparisons. Box-whisker plots shows individual tumors, median and min-max values. NS−non-significant ($p > 0.05$). **b** WT or IFNAR1 KO BM cells were transferred to congenic lethally irradiated mice. After 8 weeks mice were implanted with MC38 tumor cells and treated with ceralasertib (6.25 mg/kg b.i.d.) with or without anti-PD-L1 (10 mg/kg b.i.w.). TGR for individual mice, median, and min-max are shown. $n = 8$ per group. Statistical analysis was performed with one-way ANOVA corrected for multiple comparisons. $P$ values are shown on graphs. NS−non-significant ($p > 0.05$). **c** CD4$^+$ and CD8$^+$ T cells were isolated from spleens of naïve mice, treated for 72 h with indicated concentration of

IFN-β in the presence of CD3/CD28 antibodies. Expression of PD1 was measured by flow cytometry. Individual results ($N = 3$), mean and SEM are shown. $P$ values were calculated in One-way ANOVA with correction for multiple comparisons and shown on graphs were possible to maintain readability of the graphs **$p < 0.01$; ***$p < 0.001$. The same results were obtained after 48 h of treatment. **d** IFNAR1 KO ($N = 6$) MC38 TB mice treated for 7 days with ceralasertib as described for WT mice (Fig. S8C). Expression of PD1 was measured by flow cytometry in spleen and tumor CD4$^+$ and CD8$^+$ T cells and $P$ values were calculated in two-sided unpaired Student's t-test. Individual mice results ($N = 6$), mean and SEM are shown. **e** Tumor volume in mice reconstituted with BM from WT or IFNAR1$^{SA}$ mice and implanted 8 weeks later with MC38 tumor cells. Mice treated with 6.25 mg/kg ceralasertib for 7 days. $N = 10$ per group. Statistical analysis performed by Fishers test. Source data are provided as a Source Data file.

IFN-β combination treatment was observed in some cell lines. Collectively these data show that IFNI induction by ceralasertib can have broad impacts on the malignant and benign cell components of the TIME.

## Discussion

Until recently, the role of therapeutics targeting the DNA damage response such as PARP, ATR, DNA-PK and ATM for cancer treatment has largely focused on the tumor cells. Pre-clinically, the role of ATRi via direct targeting of tumors (predominantly in model systems without an immune component) with DDR-defects, oncogene induced replication stress, or in combination with chemotherapy, radiation, or PARP inhibitors have been well documented[1,2,10,11,16,17]. Here, we show that the ATRi ceralasertib has potential to deliver a broad and profound impact in the TIME promoting T-cell activity, and influencing the number and function of myeloid cells. This pleiotropic response was highly dependent on the dose and schedule of ceralasertib treatment. In pre-clinical models, maximal efficacy was delivered using an intermittent on-off treatment schedule, which drives anti-tumor activity in a CD8 T-cell and type I IFN dependent manner. During the on-treatment phase ceralasertib affected T-cells that display markers commonly associated with an exhausted phenotype, enabling repopulation of the TIME with T-cells with a non-activated phenotype in the off-treatment period. It was associated with up-regulation of tumor antigen specific CD8$^+$T-cells. This regimen caused up-regulation of PD-1 expression on T-cells that could potentially make these cells more susceptible to the effect of PD-L1 antibody.

Why has ceralasertib treatment improved T-cell function and priming of the TIME? In MC38 model, in addition to a direct effect on T-cells, ceralasertib had a strong effect on myeloid cells. Ceralasertib also caused activation of DCs known to promote T-cell responses and down-regulated immune suppressive myeloid cells: MDSC and TAM. Since inadequate function of DCs and immune suppressive activity of MDSC and TAM were all implicated in resistance of cancer patients to immune check-point inhibitors[28,29] it may suggest the mechanism that would be of particular clinical benefit (but not restricted to) for patients who had progressed on immunotherapy. Collectively the observations suggest that ceralasertib would have the potential to address more than one resistance mechanism in the TIME, ultimately re-invigorating the immune cell function in the tumor. This concept has been tested in human trials and in early clinical studies the combination of ceralasertib (dosed intermittently) with durvalumab has been shown to be safe and tolerable with encouraging signs of clinical efficacy in a Phase II umbrella in advanced/metastatic NSCLC patients following platinum-doublet therapy and progressed on anti-PD-(L)1-containing immunotherapy (HUDSON trial NCT03334617)[13].

Pharmacological inhibition of ATR is expected to have direct growth suppression effect on tumor cells though exacerbation of replication stress and DNA damage[1]. However, our study demonstrated that the effect of ceralasertib on the TME is not limited to

tumor cell mediated activity. Experiments in tumor-free mice and in model systems that were not sufficient to cause antitumor growth inhibitory effects as a single agent showed its ability to potentiate T-cell activation in combination with PD-L1 antibody. This implies that in the right context ceralasertib can have a profound impact on T-cell function and suggested its broad applicability in context of cancer immunotherapy (Fig. 9).

The effect of ceralasertib on several cells in the TME needs to be carefully considered. For example, sustained inhibition of ATR may have detrimental effects on T-cell numbers and function. However, the use of an intermittent ceralasertib treatment schedule resulted in reduction of T-cells displaying markers of exhaustion while allowing for T-cell recovery during the off-treatment period. This ensued increase in the population of T-cells that appear generally more reactive. Ceralasertib treatment enriched the TME for activated DCs and depleted or inactivated suppressive myeloid cells provided an environment for efficient and sustained T-cell activation. Our experiments demonstrated that the major mechanism of this effect of ceralasertib was likely mediated via up-regulation of IFNI pathway (Fig. 9). This conclusion is based on "loss of function" and "gain of function" experiments, which directly implicated IFNI as a major mechanism responsible for the effect of ceralasertib. This is consistent with previously described IFNI effects on DC and MDSC[30] Analysis of peripheral blood from patients treated with ceralasertib indicated an induction in IFNI signaling following treatment and hypothesis free unbiased analysis of the preclinical models indicate increase in IFNI related gene signatures.

Because ceralasertib impacted multiple cells in TME, it is likely, that IFN1 is produced by multiple cells, and sustained during the on-off ceralasertib treatment phases. Moreover, the induction of IFN1 is commonly seen as driving activation of immune cells, here induction of IFN1 in the context of ceralasertib treatment may also have potential to reduce tumor cell survival, alone or enhanced in combination with ceralasertib. This highlights that the combination has potential to influence therapeutic response through different but complementary mechanisms.

Two independent research group have recently reported on the schedule dependency of ATRi treatment for optimal anti-tumor activity and on the impact on growth of proliferating T-cells. An intermittent on-off ceralasertib regimen in the context of a radiation therapy combination suppressed proliferation of CD8 + T-cells on treatment which was rapidly reversed off treatment, and that ceralasertib potentiated the induction of IFN1 after radiation[31]. In a separate study Sugitani et al. reported that proliferating CD8$^+$ T-cells had hallmarks of high replication stress, exhibited marked induction of ATR expression and relied on ATR signaling to maintain viability[32]. Supplementation of exogenous thymidine to proliferating T-cells rescued ATRi induced genomic instability, cell death and IFNα/β induction in a similar fashion to cancer cells. These data are consistent with our findings and importantly demonstrates the generalizability of

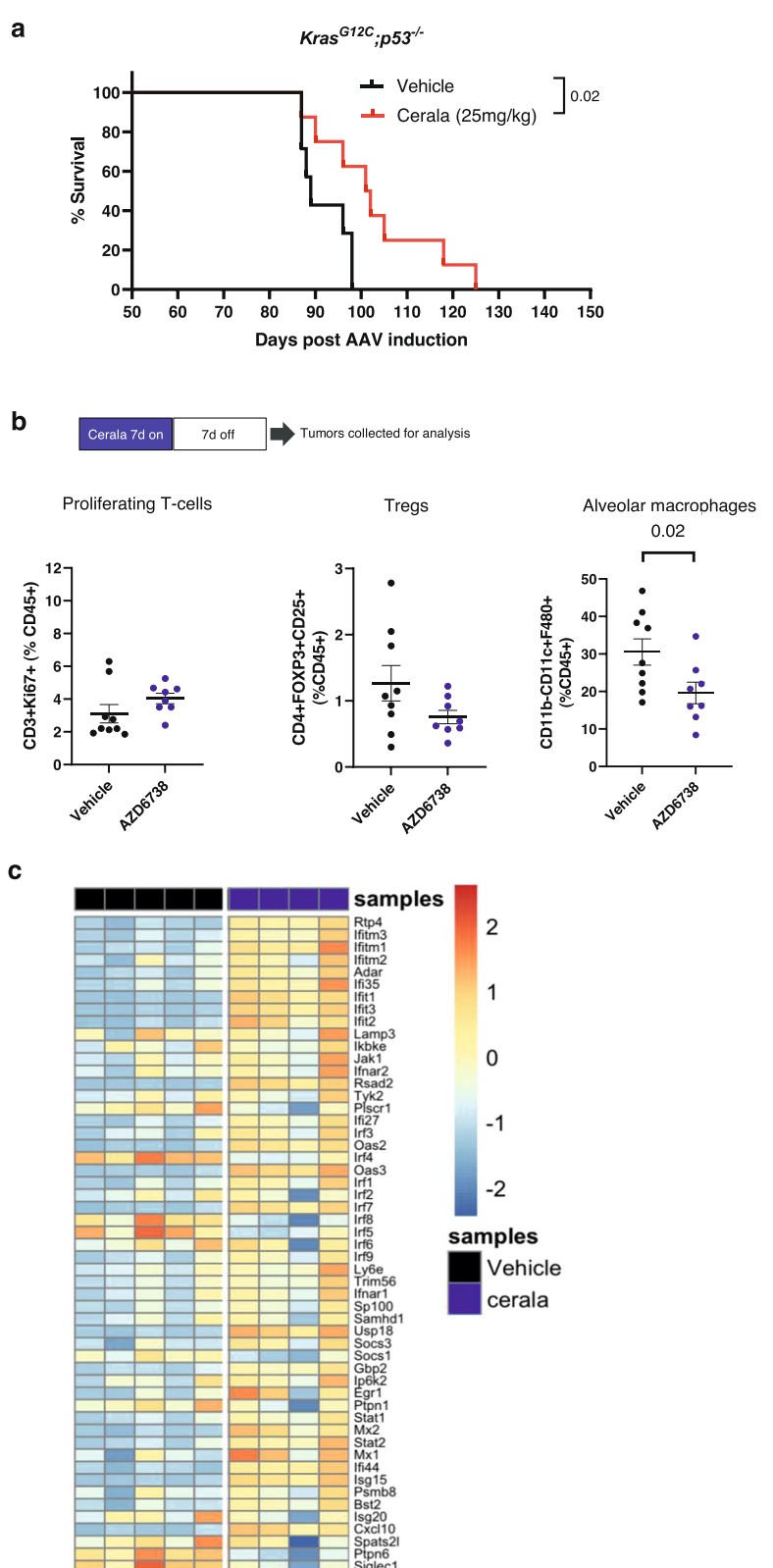

**Fig. 6 | Ceralasertib improves survival of LUAD GEMM and impacts TIME.**
**a** Kaplan–Meier survival analysis of TB mice treated with vehicle ($n = 7$) or cer-alasertib (cerala) ($n = 8$) (Day 50 7days on / 7days off + 1day on). **b** GEMM $Kras^{G12C};p53^{-/-}$ lung tumors were profiled by flow cytometry for the indicated immune populations. The results of individual mice, median and SD are shown $P$ values were calculated in two-sided unpaired Student's $t$-test. $N = 9$ for vehicle, $N = 9$

for AZD6738 groups. Non-significant ($p > 0.05$) $p$ values are not shown. $N = 5$ for vehicle and $N = 4$ for ceralasertib groups, **c** Heatmap of IFNI signature gene expression, calculated from FPKM normalized counts expressed above biological viable levels, log-transformed and normalized for gene lengths. The data was $z$-transformed to be visualized in heatmap. Source data are provided as a Source Data file.

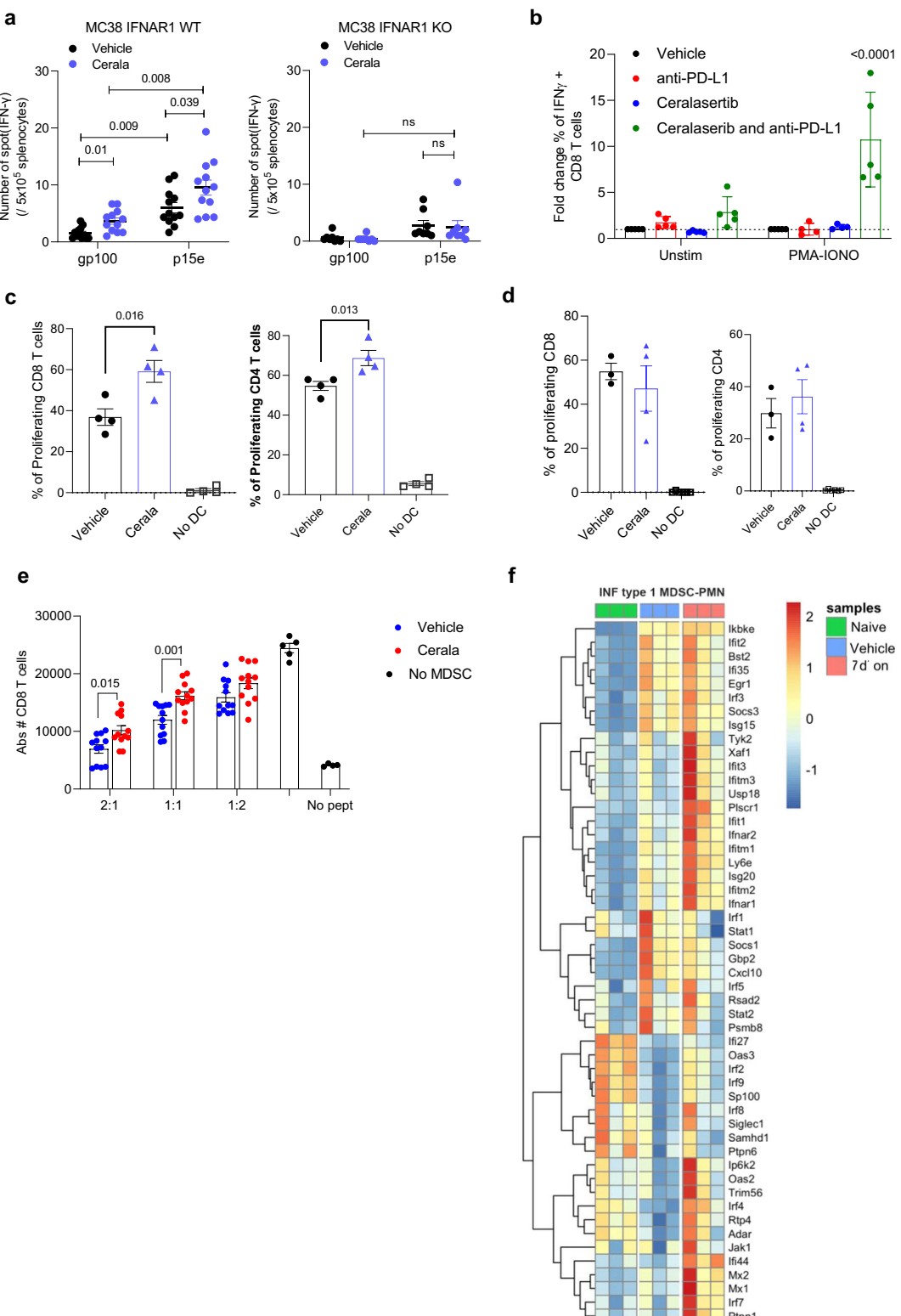

effects of ATRi to other model systems or combinations and provides a molecular basis (consistent with known ATR biology) for the observed impact on proliferating cell populations.

The broad impact of ceralasertib in the TME contrasts with studies examining the impact of PARPi in similar models[33]. Rather than inducing broad reprogramming of the TME as with ceralasertib, targeting PARP largely primed immune responses or enhanced immune checkpoint response in a tumor-intrinsic STING dependent manner[34,35].

Considering that IFNI was implicated as major mediator of the effect of ceralasertib, then what makes its effect different from IFNI inducers like STING? We believe there are several potential mechanisms that could explain this effect.

(i)   The ability of ATRi to impact T-cells and thus limit the proliferation of exhausted T-cells while preserving the pool of naïve non-activated T-cells. An intermittent dosing schedule allowed for recovery of these cells under much more permissible conditions provided by IFNI mediated changes in myeloid cells;

**Fig. 7 | IFNI signaling regulates tumor-specific T-cells. a** WT or IFNAR1 KO mice were implanted with MC38 tumor cells s.c. and treated with 25 mg/kg b.i.d. ceralasertib. Splenocytes were restimulated with tumor-specific (p15e) or irrelevant peptides (mGp100). Number of IFN-γ spots are shown. *N* = 12 WT, *N* = 6 IFNAR1 KO. Individual mouse results, mean and SEM are shown. Two-sided unpaired Student's *t*-test was used. NS−non-significant (*p* > 0.05). **b** T-cells isolated from LN of mice treated with 10 mg/kg anti-PD-L1, 25 mg/kg b.i.d. ceralasertib, or combination were plated in the presence of feeder of control splenocytes, restimulated for 6 h with cell-stimulation cocktail, and stained for intracellular IFN-γ. Fold increase of % of IFN-γ⁺ cells within CD8⁺ T-cells is shown. *N* = 5. Individual mouse results, mean and SEM are shown. One-way ANOVA with correction for multiple comparisons was used. Non-significant *p* values (*p* > 0.05) are not shown. **c** DC were isolated from LN of MC38-bearing mice (C57BL/6) treated with 25 mg/kg b.id. ceralasertib and plated with BALB/c derived splenocytes labeled with Cell Trace Far-Red at a 1:5 (DC:splenocytes) ratio. After 5 days proliferation of CD8⁺ and CD4⁺ T-cells was measured by Cell-Trace dilution. *N* = 4. Individual mouse results, mean and SEM are shown. Two-sided unpaired Student's *t*-test was usede. **d** Experiments were performed as in (**c**) except DC were isolated from LN of IFNAR1 KO MC38-bearing mice. *N* = 3−vehicle group, *N* = 4−Cerala group, *N* = 6−no DC group. Individual mouse results, mean and SEM are shown. **e** PMN-MDSC were isolated from tumors of mice treated for 7 days with 25 mg/kg b.i.d. ceralasertib and plated at different ratios with PMEL splenocytes in the presence of specific peptide. After 48 h the number of CD8⁺ T-cells was calculated by flow cytometry. *N* = 10. Individual mouse results, mean and SEM are shown. Two-sided unpaired Student's *t*-test was used. Non-significant (*p* > 0.05) *p* values are not shown. **f** IFNI signature heatmap evaluated in PMN from naïve mice (*N* = 3), tumor PMN-MDSC isolated from mice treated with vehicle (*N* = 3) or with 25 mg/kg ceralsertib for 7 days (*N* = 3). Source data are provided as a Source Data file.

(ii) ATRi can suppress proliferation of tumor cells. Although this effect was modest, it decreases the pressure from proliferative tumor cells on the TIME;

(iii) The strength and duration of the IFNI signaling could be an important factor. Indirect intermittent effect of ceralasertib on IFNI may provide more physiological regulation signal in TIME than potent stimulation with IFNI agonists.

Collectively these data establish a unique on target role for ceralasertib in activating the immune response in the TME through reprogramming of different cell types and initiating IFN mediated cross talk to different cell types in the TME.

## Methods

The study is being performed in accordance with consensus ethical principles derived from international guidelines including the Declaration of Helsinki and Council for International Organizations of Medical Sciences (CIOMS) International Ethical Guidelines, applicable International Council for Harmonization (ICH) Good Clinical Practice (GCP) Guidelines, all applicable laws and regulations, and the AstraZeneca policy on Bioethics and Human Biological Samples. The study protocol has been approved by AstraZeneca Institutional Review Board/Ethics Committee.

Murine experiments in the US were approved by the Institutional Animal Care and Use Committee of AstraZeneca (Gaithersburg, MD) and conducted in Association for Assessment and Accreditation of Laboratory Animal Care (AAALAC)−accredited and United States Department of Agriculture (USDA)−licensed facility and the AstraZeneca Global Bioethics policy. Murine studies in the UK were conducted in accordance with UK Home Office legislation, the Animal Scientific Procedures Act 1986, the AstraZeneca Global Bioethics policy or Institutional Animal Care and Use Committee guidelines. Experimental work is outlined in project license PP3292652.

### Mouse models

All mice were housed in autoclaved cages with access to food and water *ad libitum* in a sterile environment maintained with a 12 h dark/12 h light cycle at $72 \pm 2\,°F$ with $50 \pm 20\%$ room humidity. Female 8- to 10-week-old C57BL/6 and BALB/c mice were obtained from Envigo. Female 8- to 10-week-old IFNAR1 KO mice (B6(Cg)-Ifnar1tm1.2Ees/J), B6.SJL-*Ptprc*ᵃ *Pepc*ᵇ/BoyJ were obtained from The Jackson Laboratory. For studies with bone marrow chimeras, B6.SJL-*Ptprc*ᵃ *Pepc*ᵇ/BoyJ mice were irradiated at 1000 RAD and intravenously engrafted with $5 \times 10^6$ donor bone marrow cells isolated from either wild-type C57BL/6 mice, IFNAR1 KO (B6(Cg)-Ifnar1tm1.2Ees/J) or IFNAR1^SA30. Mice were observed for 8 weeks for the full reconstitution of the immune compartment before the start of the experiment. Mice that achieved >90% chimerism were used for the study. GEMM model of non-small lung (NSCLC) cancer induced in OdIN mice was described previously[27]. Male and female mice age 6-8 weeks were used in the study. This model utilizes doxocyclin inducible CAS9 mice where lung tumors are induced by exposure of lungs to AAV containing RNA CRISPR guides to delete *p53* and mutate *Kras* (*Kras*^G12C). Following viral induction mice developed lung tumor lesions within 6−7 weeks.

### Tumor cell lines and syngeneic tumor models

CT26 and MC38 colon carcinoma, A20 lymphoma and 4T1 breast carcinoma cell lines were obtained from the ATCC. MC38 were cultured in DMEM (Gibco) with 10% FBS (Gibco) and 4T1 were maintained in RMPI 1640 (Gibco) with 10% FBS at 37 °C, 5% $CO_2$. CT26 and A20 were cultured in RPMI-1640, 10% FCS, 1% L-Glutamine at 37 °C, 7.5% $CO_2$. A20 media was further supplemented with 0.05 mM 2-mercaptoethanol and 1% Pen Strep. Unless otherwise stated, mice were subcutaneously injected with $5 \times 10^5$ MC38 tumor cells in DPBS into the right flank while BALB/c mice were injected orthotopically with $1 \times 10^5$ 4T1 tumor cells in the mammary fat pad using a 27 ½g needle on Day 0 of study. For CT26 and A20, BALB/c mice were subcutaneously implanted into the right flank with $5 \times 10^5$ and $1 \times 10^7$ cells, respectively. Tumors were measured using digital calipers measured 3x a week until the end of study or when tumors reached the ethical limit of 1500−2000 mm³ using calipers. In some cases, this limit has been exceeded the last day of measurement and the mice were promptly euthanized.

All human NSCLC cell lines were originally obtained from ATCC, identities confirmed by STR, and were grown in RPMI-1640 growth media (Sigma-Aldrich) supplemented with 10% fetal bovine serum (FBS) and 2 mM glutamine at 37 °C, 5% $CO_2$.

### In vivo treatments

Ceralasertib (AZD6738)[10] was made by AstraZeneca (Cambridge, UK). Ceralasertib was formulated in 10% DMSO, 40% Propylene Glycol (Sigma-Aldrich), 50% deionized water and made fresh every 7 days. Unless indicated otherwise, starting on Day 3 post tumor cell implantation, mice were administered with ceralasertib at 6.25 mg/kg or 25 mg/kg twice daily via oral gavage on a treatment cycle of 7 days-on/7 days-off. In studies with anti-mouse PD-L1 (BioXcell, Clone 10 F.9G2), starting at Days 7−11 post tumor implantation, the monoclonal antibody was diluted in DPBS and administered intraperitoneally at 10 mg/kg every 3−4 days for a total of 4−6 doses. In CD8 depletion studies in MC38, CD8 depletion antibody (BioXCell, Clone 53−6.7) was diluted with DPBS and administered intraperitoneally at 10 mg/kg on Day −2 prior to tumor implantation, Day 0, and Day 2 post tumor implantation, and continued every 3−4 days until the end of study. In CT26 and A20, CD8 depletion antibody (BioXCell, YTS 169.4) was dosed at 10 mg/kg intraperitoneally 24 h prior to treatment start with ceralasertib and continued dosing at day 1, day2, day 8 and day 15 of ceralasertib dosing regimen. IFNAR1 blocking in CT26 was done by administering an IFNAR1 blocking antibody (BioXCell, clone MAR1-5A3) at 5 mg/kg intraperitonially 24 h prior to treatment start with ceralasertib and continued on a x3 weekly schedule for

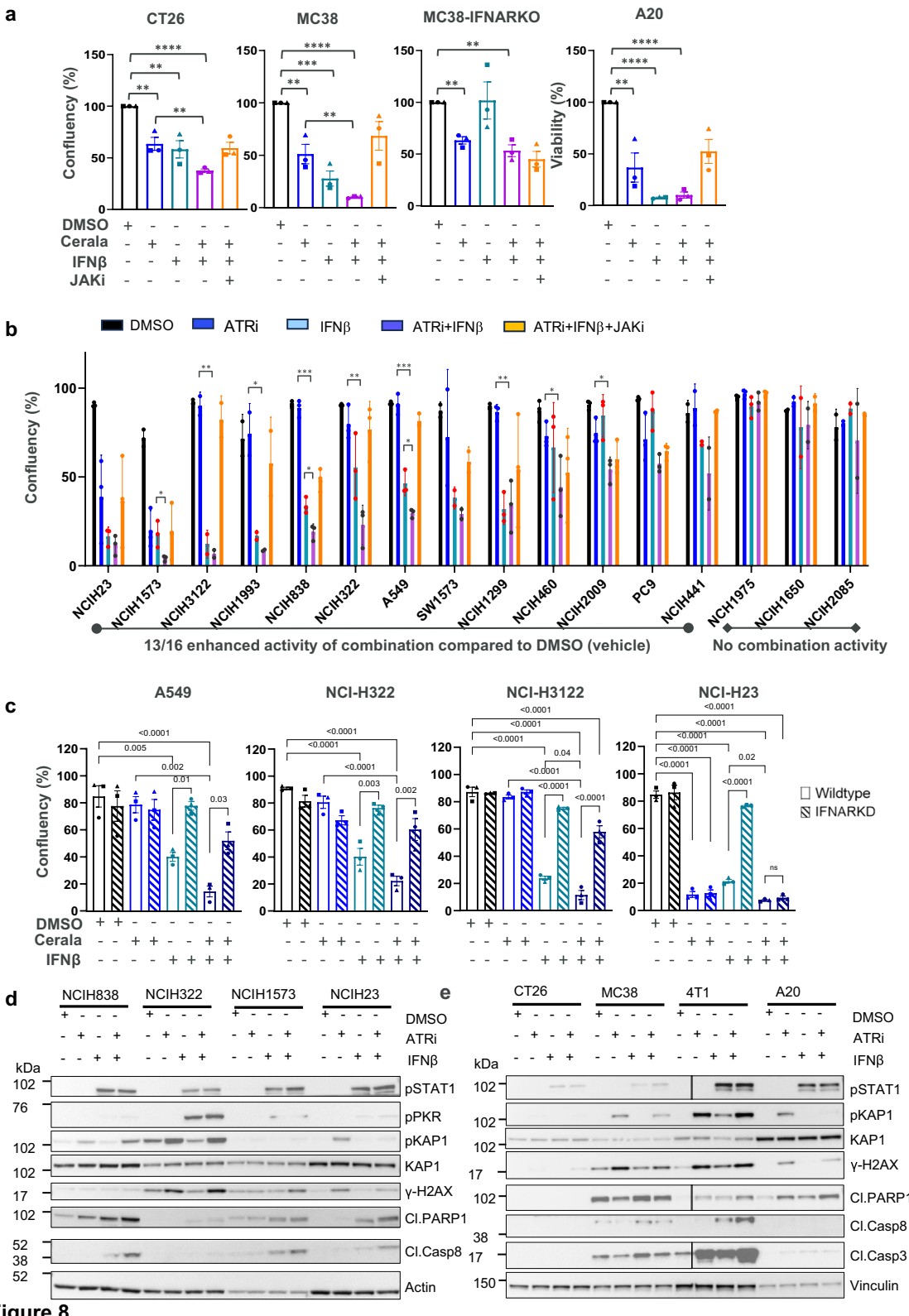

**Figure 8**

4 weeks. Control treatments vehicle were formulated 10% DMSO, 40% Propylene Glycol (Sigma-Aldrich), 50% deionized water and made fresh every 7 days and isotype antibody was diluted in DPBS and administered intraperitoneally at 10 mg/kg every 3–4 days for a total of 4 doses. Mice were marked with a subcutaneous microchip to follow tumor growth in each mouse and tumors measured with electronic caliper. Tumor volume was calculated as 0.5x Length x Width². Growth rate for each animal was calculated by fitting each tumors' growth curve to an exponential model by using the formula: $\log_{10}(\text{tumor volume}) = a + b \cdot \text{time} + \text{error}$, where a and b are parameters that corresponds to the log of initial tumor volume and growth rate, respectively. The model assumes that the error terms are normally distributed[36] using the formula: $\log_{10}(\text{tumor volume}) = a + b \cdot \text{time} + \text{error}$.

**Fig. 8 | IFN signaling enhances the response of the ceralasertib by inducing apoptosis in some cell lines. a, b** Growth inhibition activity of monotherapy ceralasertib (1 µM) or IFN-β (1 ng/ml) or the combination with and without the presence of JAK inhibitor (5 µM). **a** mouse syngeneic cancer cell lines, including isogenic MC38 (IFNAR1 intact) and MC38 IFNAR1 knockout (KO) pairs. Bars show mean and SEM of three biological replicates where each replicate is represented as a symbol. **b** 16 human NSCLC cell lines. Bars show mean and SEM for three biological replicates. Statistical analysis comparing ceralasertib vs ceralasertib+IFN-β or IFN-β vs ceralasertib+IFN-β was performed by two-sided unpaired Student's t-test and indicated on the graphs for each cell line. Asterisk are shown to maintain readability of the graphs. *$p < 0.05$, **$p < 0.01$, ***$p < 0.001$, ****$p < 0.0001$. Full

statistical comparisons are in Supplementary Fig. 17C. **c** Growth inhibition activity of ceralasertib (0.3 µM) or IFN-β (1 ng/ml) or the combination in parental NSCLC cell lines with IFNAR1 (Wild-type) compared toIFNAR1 siRNA knockdown (IFNAR KD) cells. Bars show mean and SEM of three biological replicates where each replicate is represented as a symbol. Statistical analysis was performed by two-sided unpaired Student's t-test. Asterisk are shown to maintain readability of the graphs. *$p < 0.05$, **$p < 0.01$, ***$p < 0.001$, ****$p < 0.0001$. Western blot assessing target engagement and apoptosis markers following 24 h of indicated treatments in (**d**), human NSCLC cell lines and (**e**), mouse cancer syngeneic cell lines. Two experiments with the same results were performed. Source data are provided as a Source Data file.

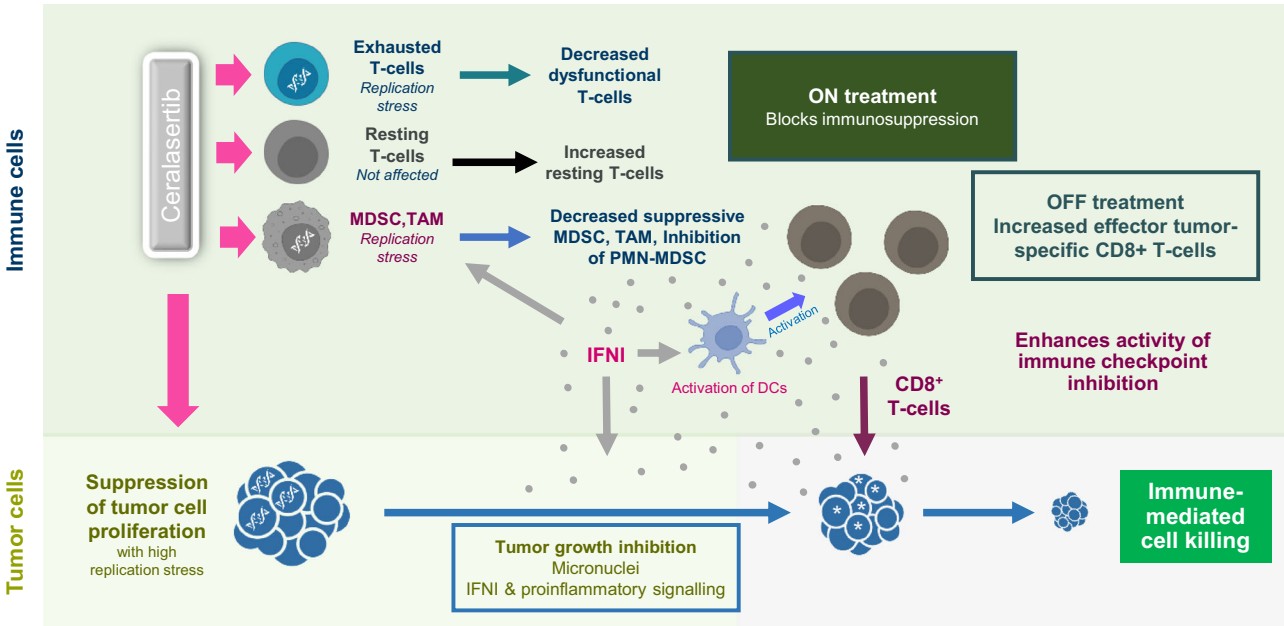

**Fig. 9 | Ceralasertib treatment schedule results in diverse effects within the tumor microenvironment.** Ceralasertib treatment reduces tumor cell proliferation and depletes exhausted CD8+ T cells, M-MDSC and TAM by inhibiting ATR function in cells with replication stress. Treatment also induces type I IFN that neutralizes suppressive activity of M-MDSC and PMN-MDSC, activates DCs, upregulate PD-1 expression on T cells, and enhances the inhibitory effect of ATRi on tumor cell proliferation. Intermittent scheduling of ceralasertib allows for

appearance of newly generated functionally competent T cells. These T cells enter tumor microenvironment with activated of DCs, decreased presence of immune suppressive myeloid cells, and less potently proliferating tumor cells. Combination of these factors makes T cells more effective in recognizing tumor antigens. Increased PD-1 expression makes these T cells more susceptible to check-point inhibitors, which result in potent antitumor effect.

## INSPECT in vivo tumor growth rate analysis

Using INSPECTumorTumors (IN-vivo reSPonsE Classification of Tumors) R package [https://CRAN.R-project.org/package=INSPECTumors] the individual animal's response to a treatment were classified as either non-responder, modest-responder, stable-responder or regressing-responder. Then a Bayesian ordered logistic regression was used to assess whether the treatment led to a significant change in the classifications. In summary, control data is first used to establish a 95% prediction interval for normal growth by fitting a Bayesian mixed effect model to the log transformed growth data. Then individual animals are assessed and identified as being responders if 3 or more consecutive readings sit below the 95% normal prediction window. These responders are then further classified by calculating the animal's individual growth rate and depending on the growth rate observed are classified as either modest, stable or regressing responders.

## Suppression assay

MDSC were isolated from tumors after 7 days of treatment with Ceralasertib with the last dose administered 2 h before taking down. Tumors were digested using mouse tumor dissociation kit (Milteniy) according to manufacturer instructions. MDSC were isolated from

tumor cell suspension as previously described[37]. Briefly, PMN-MDSC were isolated after Fc Blocking (2.4g2 Ab, BD bioscience) using anti-Ly6G Micobeads UltraPure (Miltenyi Biotec). M-MDSC and TAMs were Isolated in 2 steps: 1) CD45 enrichment with CD45 (TIL) Microbeads, Mouse (Miltenyi Biotec) followed by Sorting with a FUSION or ARIA II cell Sorter (BD Bioscience). For the sorting enriched CD45+ cells were Fc-Blocked using anti mouse CD16-32 (2.4g2 BD Bioscience) then staining with: Aqua L/D (Thermo) CD11b-BV421, Ly6G-FITC, Ly6C-APC, F4/80-PE (Biolegend). M-MDSC were isolated as L/D⁻CD11b⁺Ly6G⁻Ly6C^{high}F4/80⁻ while TAM L/D⁻CD11b⁺Ly6G⁻Ly6C^{int/low}F4/80⁺. Myeloid cells were plated at different ration with Cell Trace labeled−PMEL splenocytes (50,000) in the presence of mGP100 peptide (ANASPEC) at 1 ng/ml for 48 h. After 48 h live CD8 T-cells were counted by flow cytometry.

## Ex vivo DC function and MLR

Mice (WT or IFNAR1 KO) were implanted on both flanks with MC38 tumor cells and 14 days later were treated for 7 days with ceralasertib. Tumor draining LN (inguinal) were collected and DC isolated as previously described[38]. DC were plated in 96-well U-bottom plates with 50,000 Cell Trace Far Red (ThermoFisher) labeled splenocytes from

naïve BALB/c mice in a 1:5 ratio with DC. After 5 days cells were stained for Aqua L/D, CD3, CD4 and CD8 and Cell Trace dilution assessed for each T-cell subset. The proportion of proliferation was calculated as % of CellTrace low for each subset.

## In vitro DC differentiation and T-cell activation

DC were generated by plating $10^5$ bone marrow cells/well from WT or IFNAR1 KO with GM-CSF (20 ng/ml R&D System) + IL-4 (10 ng/ml R&D System). On day 3 and 5, half of medium were replaced with fresh medium including GM-CSF (40 ng/ml) +IL-4 (20 ng/ml). On day five, cell were treated with LPS (100 ng/ml, Sigma-Aldrich) or ceralasertib for two more days. On day 7, cells were harvested for assessment of activation markers expression by flow cytometry or DC isolation using CD11c UltraPure, mouse (Miltenyi).

## T-cell activation with BM derived DC

T-cells were isolated from spleen of PMEL mice using EasySep™ Mouse T-cell Isolation Kit (Stemcell) and stained with CellTrace Far Red Cell Proliferation Kit (Thermofisher). Stained T-cells were co-cultured with isolated BM-derived DCs with mgp100 peptide (1 ug/ml, Anaspec). Two days after co-culture, the cells were stained for surface markers and at last step, same volume of CountBright™ Absolute Counting Beads (Thermofisher) were added to each well analyzed by flow cytometry.

## ELISPOT for mouse IFN-gamma

Mouse IFN-γ ELISPOT kit (BD) was used for the experiment. ELISPOT plate was coated with purified anti-mouse IFN-γ (5 μg/ml) overnight at + 4 °C. The coated plate was washed and blocked with RPMI-1640 medium (with 25 mM HEPES and L-glutamine and 10%FBS + 1%P/S). $5 \times 10^5$ Splenocytes from WT or IFNAR1 KO mice were seeded in each well of the coated plate with endogenous tumor peptide (p15e-KSPWFTTL 1 ug/ml, MBL), or irrelevant peptide (mgp100 1 ug/ml, Anaspec), overnight at +37 °C. On the following day, the plate was washed sequentially and incubated with biotinylated anti-mouse IFN-γ (2 ug/ml) than with streptavidin-HRP and AEC Substrate Set (BD). The dots on dried well were counted using ImmunoSpot Analyzers (CTL).

## Mouse RNAseq

RNA was extracted using Quick-RNA Microprep Kit (Zymo Research). Libraries for RNA sequencing were generated using the Takara SMART-Seq v4 PLUS Kit (Takara Bio; R400752) according to the manufacturer's protocol, with the following specificities. An input of 10 ng of RNA per sample calculated from Agilent TapeStation DV200 values was used for cDNA generation. Post cDNA generation cleanup, cDNA for all samples were normalized to the cDNA concentration of the lowest yielding sample. Thus, cDNA input into library preparation was the same for all samples. Final libraries were assayed with Agilent TapeStation D5000 to determine quality and average fragment lengths. Final libraries were then assayed with KAPA Library Quantification Kit to determine final concentrations. Library sequencing was completed on an Illumina NovaSeq 6000 with a target pair-end read count of 100 M per sample.

## RNAseq

Transcripts counts were analysed with EdgeR[39] to normalize the data and quantify differential expression. Data normalization was performed using the relative log expression (RLE) implemented in EdgeR. After normalization, transcripts abundance was filtered for low and no reads. We retained transcripts for which counts per million (cpm) were greater or equal than 2 in at least two samples. Transcripts sequenced in each group were compared to one another to evaluate transcriptome coverage across our experimental conditions. For each comparison, we evaluated the biological variation using the maximization of the negative binomial dispersion using the empirical Bayes likelihood function, as implemented in EdgeR. We also fitted a generalized linear model to de-confound the data for the effects of covariates. like sample type and different treatments. Differentially expressed transcript genes were computed for fold change FC > 2 FDR < 0.01 and p-value $p < 0.01$, which were evaluated using the quasi-likelihood (QL) methods with empirical-Bayes Test in EdgeR. Differentially-expressed transcripts were annotated using BioMart[40] based on their Ensembl transcript Id to recover gene symbol, gene description and biotype. Heatmaps were evaluated on FPKM counts ain log scale after transcripts were normalized for gene lengths.

## Mouse flow cytometry

Single cell suspension from disaggregated tumor, spleen and bone marrow cells were prepared and red blood cells lysed using ACK lysing buffer (Life Technologies). Cells were incubated with Fc-block for 10 min at 40 C followed by staining with a cocktail of MAbs and acquired for FACS on a BD Symphony flow cytometer. Results were analyzed using FlowJo version 10.7.1 software. The list of antibodies used is provided in Supplementary Table 1.

## T-cell Restimulation from MC38

T-cells were isolated from draining LN of MC38-bearing mice treated for 7 days with Ceralasertib using easySep Mouse T-cell Isolation kit (STEMCELL) and plated overnight in complete medium + naïve splenocytes in a 1:1 ratio. During the last 6 h of culture Cell Stimulation cocktail plus protein transport inhibitors was added (ThermoFisher). Control wells were treated with protein transport Inhibitors only. After 6 h cells were stained for Aqua/L/D, CD3 and CD8, fixed and permeabilized using eBioscience FOXP3/Transcription factor staining buffer set (ThermoFisher) and intracellular stained for IFN-γ. Cells were analyzed by Flow cytometry.

## Human T-cell isolation and in vitro CTV proliferation assays

Human peripheral blood mononuclear cells (PBMCs) were isolated from leukocyte cones, supplied by NHS Blood and Transplant Service (NHSBT, UK) as anonymized samples from consenting donors. Lymphoprep, magnetic selection kits and cytokines were purchased from Stemcell Technologies. Following density centrifugation, CD3+ T-cell subtypes and CD14+ monocytes were isolated by magnetic selection and cultured in RPMI (Sigma R8758-500 mL, with 10% FBS) at 37 °C and 5% $CO_2$. Immune cells were labeled with CellTrace™ Violet (Thermo Fisher (TF) C34557) and plated at $0.5-2 \times 10^6$ per mL. T cells were cultured in 96-well round-bottomed plates, with or without the addition of anti-CD3/CD28 Dyna beads at ratio of 1:1 (TF 11131D). Following treatment with DMSO or ceralasertib, cultures were dissociated (TF 13151014), stained for viability (TF L34976), surface/intracellular markers (Biolegend) and fixed for analysis by flow cytometry (BD FACSCelesta).

## Western blot analysis and antibodies

For in vitro cell cultures, cells were collected and lysed in RIPA buffer (TF 89901; with complete protease and phosphatase inhibitors (Sigma). For mouse tumor tissue, tissue was homogenized in Tris buffer containing 10% glycerol, sodium orthovanadate and sodium-fluoride, and supplemented with phosphatase and protease inhibitor. Protein lysates were normalized for protein content (Pierce #23227) and boiled in LDS/reducing agent (TF NP0007/NP0004) before loading 20 μg onto 4–12% BIS-TRIS gels (Invitrogen) for SDS-PAGE. Gels were transferred to nitrocellulose using iBlot2 (TF) and probed with antibodies from Cell Signaling Technology and Abcam. Immunostaining was visualized by HRP-labeled secondary antibodies (Jackson ImmunoResearch) and enhanced chemiluminescence (Thermo Fisher #34075) using a 16 bit CCD camera (Syngene GBOX), quantified by densitometry (ImageJ).

Antibodies: Phospho-ATM Ser1981 (Abcam ab81292), Total ATM (Abcam ab78), Phospho-ATR Ser1989 (GeneTex GTX128145), Total-ATR (Abcam ab4471), Phospho-H2AX Ser139 (CST #9718), Phospho-RPA2 Ser4/8 (Bethyl laboratories A300 245 A), total RPA (Bethyl laboratories A300 244 A), Phoshpo-CHK2 Thr68 (CST#2661), Phospho-KAP1 Ser824 (Abcam ab70369), p53 (CST#9282), Histone H3 (CST #3638), CHK1 (CST#2360), Phospho-PKR Thr446 (Abcam ab32036), Cleaved Caspase-8 Asp387 (Mouse Specific CST#8592), Cleaved PARP Asp214 (Mouse Specific CST#9548, Cleaved Caspase-3 Asp175 (CST#9661), Cleaved PARP Asp214 (Human Specific CST#9541), β-Actin (CST#4967), Vinculin E1E9V (CST#13901).

## Mouse mass cytometry (CyTOF)

Single-cell tumor suspensions were prepared using the mouse tumor dissociation kit (Miltenyi Biotec; #130-096-730) as per manufactuer's instructions. $3 \times 10^6$ cells from each sample were stained with 5 μmol/L Cell-ID Cisplatin (Fluidigm; #201064) at room temperature for 1 min in MaxPar PBS (Fluidigm; #201058) and were barcoded using the Cell-ID 20-Plex Pd Barcoding Kit (Fluidigm; #201060) as per manufacturer's instructions. Cells were fixed in 1x Fix/Perm solution (Thermo Fisher; #00-5123-43, #00-5223-56) at 4oC for 15 min then blocked in anti-CD16/CD32 antibody (Thermo Fisher; #14-0161-86), and stained at 4 °C overnight with fluorophore- or metal-conjugated antibodies in 1x Permeabilization Buffer (Thermo Fisher; #00-8333-56). Cells were washed twice and stained using metal-conjugated secondary antibodies at 4 °C for 30 min then washed twice and stored in MaxPar Fix and Perm Buffer (Fluidigm; #201067) with 125 nmol/L Cell-ID Intercalator-Ir (Fluidigm; #201192 A). The list of antibodies used is provided in Supplementary Table 2. Immediately prior to acquisition, samples were washed twice using Cell Staining Buffer (Fluidigm; #201068), twice with MaxPar Cell Acquisition Solution (Fluidigm; #201241) and pellets resuspended in MaxPar Cell Acquisition Solution then filtered twice through 70-μm strainer (Greiner; #542070) before acquisition on the Helios CyTOF System (Fluidigm). Spectral overlap compensation was performed by staining anti-mouse or anti-rat/hamster beads (BD Biosciences; #552843, #552845) individually with each of the metal-conjugated antibodies. For antibodies raised in rabbit or goat, anti-mouse beads were first coated with mouse anti-rabbit IgG or mouse anti-goat IgG (Thermo Fisher; #31213 or #31107), washed and then incubated with rabbit or goat metal–conjugated antibody. Antibodies were not available as metal conjugates were labeled using Maxpar X8 Antibody Labeling Kits (Fluidigm; #201141A–201156 A, #201158A–201176 A; Sigma, #203440) as per manufacturer's instructions or as described elsewhere[41]. Sample de-barcoding was performed using CyTOF Software (Fluidigm) or manual gating. Data were analyzed using FlowJo software (V.10, Treestar) or Cytobank. Data compensation was performed using CATALYST package[42]. Cell clustering was performed with FlowSOM[43] and ConsensusClusterPlus using 'cluster' function in CATALYST package. The identified clusters were visualized using tSNE based on cytofWorkflow (https://github.com/markrobinsonuzh/cytofWorkflow). Statistical significance of differences in cell frequency were processed with a Beta Regression[44].

## Mouse bulk RNA sequencing (RNA-seq) from CT26

CT26 tumor tissues were isolated and total RNA was extracted using the RNeasy 96 QIAcube HT Kit (Qiagen; C/N 74171) according to manufactuers guidelines with a DNase digest included. RNA concentration was determined by Qubit Flex Fluorometer (Invitrogen), RNA purity was determined using a NanoDrop Eight (Thermo Scientific) and RNA integrity was measured using a 4200 Tapestation (Agilent). All samples had a RINe of ≥7.0. Libraries were prepared using NEBNext Ultra II Directional RNA Library Prep Kit for Illumina (New England BioLabs; #E7760L) as per manufactuer's guidelines with 1000 ng of RNA input. Ribosomal RNA was removed using the NEB-Next® Poly(A) mRNA Magnetic Isolation Module (New England

BioLabs; E7490L). Libraries were subjected to 9 cycles of PCR with unique dual indexes (New England BioLabs; E6440L). Libraries were subsequently quantified by Qubit Flex Fluorometer (Invitrogen), and library sizes were determined by 4200 Tapestation (Agilent) and pooled equimolar. Each library was loaded onto one lane of an S4 v1.5 flow cell (300 cycles) (Illumina, #20028312) on an Illumina NovaSeq 6000 at $2 \times 150$ paired-end configuration.

## Mouse single cell RNA sequencing (ScRNA-seq)

CT26 tumor tissues were chopped and transferred into a tube containing RPMI (Thermo Fisher, Gibco 61870-044). Single-cell suspension was prepared by treating tumors with a mix of enzymes using mouse tumor dissociation kit (Miltenyi Biotec; No. 130-096-730) for 40 min at 37 °C with agitation. Cell suspensions were filtered using 70 μM strainer (Greiner #542070) and washed with 10 ml RPMI then centrifuged at 13000 rpm 5 min at 4 °C. The antibodies (see below) were diluted in 50% flow cytometry staining buffer (Thermofisher #00-4222-26) plus 2 mM EDTA with 50% Brilliant Stain Buffer (BD Biosciences #563794). Cells were blocked in anti-CD16/CD32 antibody at 1:100 (ThermoFisher#14-0161-86) then stained for 30 min on ice in the dark. Cells were washed twice in flow cytometry staining buffer (Thermofisher #00-4222-26) plus 2 mM EDTA and resuspended in flow buffer plus 2 mM EDTA with 1:3000 DAPI just prior to sorting. Compensation beads (Thermofisher 01-2222-42) and FMO CD3 stained tumor sample were also stained. The flow sorter (BD Aria) was set upstream stabilized and then compensated, and purity check performed. T-cell collection gate (50,000 CD3$^+$ CD11b -DAPI$^-$ singlets) was setup and report CD4$^+$ and CD8$^+$ proportions were also recorded. Cells were sorted into 15 ml falcon collection tube containing 8 ml RPMI +glutamax + 2% 0.22 micron filtered heat inactivated FCS. The cells post sort were centrifuged at 1500 rpm (<300 g), 5 min 4 °C. Majority of media was aspirated, and cells washed in cold 9 ml PBS (Sigma # D8537) / 0.04% BSA (RNase free, 0.22 micron filtered Sigma # B6917) per tube. The wash was aspirated and the cells per tube counted using haemocytometer and trypan blue.The cells were kept on ice and resuspended so 400c/ul in PBS /0.04% BSA. The protocol for the Chromium Next GEM Single Cell 5' Kit v2, 16 rxns (#1000263) was then followed as per manufacturer guidelines.

Flow staining cocktail (antibody-fluorochrome): CD45-BV786 (BD #564225), CD3-BUV395 (BD#563565), CD4-BV711 (Biolegend#100557), CD8a-BV650 (Biolegend#100742), CD25-PE/Cy7 (Biolegend#102016), CD11b-Pe CF594 (BD#562287), DAPI (Biolegend#422801).

## Mice and LCMV clone 13 infections

Five-week-old female C57BL/6 J (B6J) mice were rested 1-week prior to LCMV Clone 13 (C13) infection. Virus stocks of the LCMV C13 were plaque purified on Vero E6 cells, grown in BHK-21 cells, and quantified and qualified by plaque assay and in vivo phenotypic profiling of INFγ, TNFα and IL-2 after a 5-h restimunlation with gp33, 9-mer peptide. Mice were infected intravenously (i.v.) with $2 \times 0^6$ PFU/500 mL of LCMV C13. C13 infected mice were randomized on day 10 based on body weights which ranged 12.6–14.1 grams. Mice weighing more than 14.1 grams were excluded from the C13 infected groups. LCMV Plaque Assay. Vero E6 cells were plated at $3.5 \times 10^5$ cells per well in a 6-well plate 1 day prior to infection at 37 °C and 5% CO$_2$. Viral stocks were diluted 1/10 and 200 μL sample was overlayed per well and gently hand rocked every 15 min 1.5 h. Virus overlay was removed and 1% methylcellulose in 1XDMEM was overlayed for four days. Methylcellulose was removed and monolayer was fixed and permeabilized with 4% PFA and 0.25% TritonX-100 for 20 min at RT. Anti-LCMV NP mAb (BioXcell, Lebanon, NH) at 5ug/mL concentration (2.5% BSA in 1XPBS) for 2-h at RT followed by donkey anti-mouse HRP (ThermoFisher, Carlsbad, CA) to each well and incubate at RT for 1-h. Monolayer was developed with AEC solution (BD Bioscience, Franklin Lakes, NJ) and incubate ≈10–15 min at RT. Plaques were than enumerated and PFU's were calculated.

**In vitro Incucyte proliferation and viability cell assay in combination with IFN-β, JAKi and IFNAR1 siRNA knockdown**

In vitro cell line cultures were dosed with ceralasertib in 96 well plates at clinically relevant doses of 0.3 μM or 1 μM. In some conditions recombinant 1 ng/ml Human or mouse IFN-β were used and were obtained from Peprotech (300-02BC) and R&D Systems (8234-MB-010/CF), respectively. In some conditions Ruxolitinib (JAK inhibitor; JAKi) was used and obtained from Sigma-Aldrich (S1378). For suspension cell lines cell viability was measured after 6 days using CellTiter-Glo as per manufacturer's instructions (Promega, Madison, WI, USA; G7570). For adherent cell lines live-cell imaging was acquired at 10x magnification at every 4 h and cell confluency were quantified using Incucyte ZOOM 2018A software (Essen Bioscience). Cell growth was quantified using percentage phase confluency. Cell growth inhibition for each cell line was obtained at the timepoint where the DMSO confluency was around 90%.

In some conditions, IFNAR knockdown cells were generated by preparing a mix of OptiMEM (ThermoFisher 31985070) with siRNA IFNAR (5 nM) (ThermoFisher AM16708). Negative Control siRNA (ThermoFisher 4390843) was used as a control. Following treatment with siRNA for 2 days, cultures were dissociated and stained for viability using LIVE/DEAD cell stain (ThermoFisher L34976) and surface marker IFNAR antibody (ThermoFisher MA524850) and secondary AF-647 conjugated antibody (ThermoFisher A32733) and assessed for receptor expression by flow cytometry (BD Fortessa).

**Statistical analysis**

Unless otherwise stated, one-way ANOVA with Tukey's multiple comparisons test was used to determine statistical differences between treatment groups (GraphPad Prism). For data containing only two treatment groups, a student's unpaired $t$-test was used. A value of $p \leq 0.05$ was considered statistically significant (GraphPad Prism) with values noted in individual figures or figure legends.

**Reporting summary**

Further information on research design is available in the Nature Portfolio Reporting Summary linked to this article.

## Data availability

Raw data for sequencing are deposited Array express with links provided below: https://www.ebi.ac.uk/biostudies/arrayexpress/studies/E-MTAB-13703. https://www.ebi.ac.uk/biostudies/arrayexpress/studies/E-MTAB-13704. https://www.ebi.ac.uk/biostudies/arrayexpress/studies/E-MTAB-13707 https://www.ebi.ac.uk/biostudies/arrayexpress/studies/E-MTAB-13713. The remaining data are available within the Article, Supplementary Information or Source Data file. Source data have been provided with this paper. Source data are provided with this paper.

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

## Acknowledgements

The authors would like to acknowledge the in vivo teams (LAS and TDE Bioscience AstraZeneca, UK and ICC Bioscience, USA) for supporting the mouse studies.

## Author contributions

Conceptualized, designed, and the overall supervised the study, wrote the manuscript—AL, DIG, and STB. Designed and performed the specific experiments, wrote the manuscript—ELH, ES, AK, and DT. Analyzed data, edited the manuscript—SF, MC, and ED. Performed experiments and/or analyzed the data—MMi, CM, AH, MMa, MK, AS, LM, RM, EC, SEC, JCB, GK, NS, MS, SG, AP, STa, STu, HF, AM, KM, CL, SH, DS, LC, ZW, DC, JB, AL, FJCN, LN, CPM, TB, GSA, ZR, AB, BP, GNJ, AJP, VVA, and SI. Provided material, edited manuscript—SYF. Provided material, analyzed data UD, PJL, MAN, and CAGR.

## Competing interests

S.Y.F., U.D., P.J.L., M.A.N. and C.A.G.R. declare no competing interests. All other authors are employee and shareholders of AstraZeneca.

## Additional information

Elizabeth L. Hardaker[1,16], Emilio Sanseviero[2,16], Ankur Karmokar[1,16], Devon Taylor[2,16], Marta Milo[1], Chrysis Michaloglou[1], Adina Hughes[1], Mimi Mai[2], Matthew King[1], Anisha Solanki[1], Lukasz Magiera[1], Ricardo Miragaia[1], Gozde Kar[1], Nathan Standifer[2,15], Michael Surace[2], Shaan Gill[1], Alison Peter[1], Sara Talbot[1], Sehmus Tohumeken[2], Henderson Fryer[2],

Ali Mostafa[2], Kathy Mulgrew[2], Carolyn Lam[1], Scott Hoffmann [3], Daniel Sutton[3], Larissa Carnevalli [1], Fernando J. Calero-Nieto [1], Gemma N. Jones[1], Andrew J. Pierce [1,14], Zena Wilson[1], David Campbell[2], Lynet Nyoni [1], Carla P. Martins[1], Tamara Baker[4], Gilberto Serrano de Almeida[4], Zainab Ramlaoui [2], Abdel Bidar[5], Benjamin Phillips[6], Joseph Boland[2], Sonia Iyer [7], J. Carl Barrett[2], Arsene-Bienvenu Loembé[8], Serge Y. Fuchs [9], Umamaheswar Duvvuri[10], Pei-Jen Lou [11], Melonie A. Nance[12], Carlos Alberto Gomez Roca [13], Elaine Cadogan [1], Susan E. Critichlow[1], Steven Fawell[7], Mark Cobbold[2], Emma Dean [1], Viia Valge-Archer[1], Alan Lau [1], Dmitry I. Gabrilovich [2,17] ✉ & Simon T. Barry [1,17] ✉

[1]Oncology R&D, AstraZeneca, Cambridge, UK. [2]Oncology R&D, AstraZeneca, Gaithersburg, MD 20878, USA. [3]Imaging and Data Analytics, AstraZeneca, Cambridge, UK. [4]CPSS AST, AstraZeneca, Cambridge, UK. [5]CPSS, Imaging, AstraZeneca, Gothenburg, Sweden. [6]Data Sciences & Quantitative Biology, Discovery Sciences, R&D, AstraZeneca, Cambridge, UK. [7]Oncology R&D, AstraZeneca, Boston, MA, USA. [8]Early Clinical Development, AstraZeneca, Oss, the Netherlands. [9]Department of Biomedical Sciences, School of Veterinary Medicine University of Pennsylvania, Philadelphia, PA 19104, USA. [10]UPMC Department of Otolaryngology and UPMC Hillman Cancer Center, 200 Lothrop St. Suite 500, Pittsburg, PA 15213, USA. [11]National Taiwan University Hospital, No. 7, Chung Shan S. Rd. (Zhongshan S. Rd.), Zhongzheng Dist., Taipei City 10002, Taiwan. [12]VA Pittsburgh Healthcare System, University Drive C, Pittsburg, PA 15240, USA. [13]Institut Claudius Regaud-Cancer Comprehensive Center, 1 Avenue Irene Joliot-Curie, IUCT-O, Toulouse 31059 Cedex 9, France. [14]Crescendo Biologics Limited, Cambridge, UK. [15]Present address: Tempest Therapeutics, Brisbane, CA, USA. [16]These authors contributed equally: Elizabeth L. Hardaker, Emilio Sanseviero, Ankur Karmokar, Devon Taylor. [17]These authors jointly supervised this work: Dmitry I. Gabrilovich, Simon T. Barry.
✉e-mail: dmitry.gabrilovich@astrazeneca.com; simon.t.barry@astrazeneca.com

