## [Peer Review File · Nature Communications]

The ATR inhibitor ceralasertib potentiates cancer checkpoint immunotherapy by regulating the tumor microenvironmentEditorial Note: Parts of this Peer Review File have been redacted as indicated to maintain patient confidentiality.

REVIEWER COMMENTS

Reviewer #1 (Remarks to the Author): with expertise in ATR

Hardaker et al. show that ATRi AZD6738 modulates the tumor microenvironment and potentiates responses to checkpoint blockade. The paper is a true tour de force and will be highly impactful, particularly if published with the manuscript describing results from the HUDSON trial in NSCLC (I am not a reviewer of this clinical paper).

Two great strengths of the paper are the use of both preclinical mouse models and human specimens, and the analyses of the impact of Ceralasertib in different cell types. The ATR kinase inhibitors are a very promising class of drugs. But as this paper points out, schedule is absolutely critical.

I would argue that this manuscript sets a new standard for publication of responses to DNA damage response inhibitors in vivo. It has immediate clinical implications for a large number (10's) of ongoing clinical trials of ATR kinase inhibitors, both industry and NCI ETCN supported.

Fundamental advances documented in this manuscript:

Comprehensive preclinical assessment of Ceralasertib plus PD-L1 antibody treatment shows that the combination increases survival (Figure 1b is highly impactful) and that increased survival is CD8+ T cell-dependent in preclinical animal models. Analyses of specimens from [REDACTED] shows that Ceralasertib as monotherapy decreases CD8+ T cell proliferation in patients. Taken together, these data confirm that intermittent treatment with Ceralasertib is a critical consideration for response. These data are impactful for studies with Ceralasertib and at least four other ATR kinase inhibitors that are in the clinic.

CyTOF analyses of tumors from preclinical models treated with Ceralasertib show a reduction in exhausted CD8+ T cells. This is original and highly significant and impactful. It may well be that Ceralasertib has a role in re-sensitizing tumors to immune checkpoint blockade, both refractory and previously treated that have acquired resistance.

Analyses of specimens from [REDACTED] shows that Ceralasertib as monotherapy induces IFN1 in patients. Comprehensive RNA-seq analyses of CT26 tumors and patient PBMCs reveals the induction of IFN1. While this has been shown before, the authors advance the observations with mechanistic studies. IFNAR1 blocking antibody attenuated the effect of Ceralasertib in preclinical models. Bone marrow transplant with IFNAR KO marrow attenuated the effect of Ceralasertib in preclinical models. This, IFN1 signaling is critical for response to Ceralasertib. This has not been shown previously in preclinical models.

Ceralasertib activates dendritic cells in preclinical models. Ceralasertib depletes MDSCs in preclinical models. These findings advance the mechanism described in the previous paragraph insofar as they start to tease out which cells are critical for the response.

The work presented supports the conclusions and claims. I identified no flaws in the data analysis, interpretation and conclusions. The methodology is sound and the work exceeds expected standards in the field. There is enough detail provided in the methods for the work to be reproduced.

Minor weaknesses:

Figure 2e: I can't find the time point when activated CD8+ T cells were harvested. The ATM result is curious. I have not seen ATM phosphorylation and higher ATM expression in naïve CD8+ T cells than proliferating CD8+ T cells before. This result is different to that described by Sugitani et al. Cell Reports 2022. The authors should comment on this.

CT26/BALBc and MC38/C57BL6 are used to model T cell and myeloid responses to Ceralasertib. If the myeloid analyses were performed in both models, it would be good to show the CT26/BALBc data in the supplement. I think it's good to show that the impact of a pharmacologic on the immune system is different in different mouse models, as it almost certainly will be in humans.

Vendetti et al. JCI Insight, 8(4):e165615 (2023) demonstrated that the "Schedule of ATR inhibitor AZD6738 potentiates or abolishes anti-tumor CD8+ T cell responses to radiotherapy." This manuscript also showed that intermittent Ceralasertib suppressed proliferation of CD8+ T cells on treatment which was rapidly reversed off treatment and that Ceralasertib potentiates the induction of IFN1 after radiation.

Sugitani et al. Cell Reports, 20, 40(12):111371 (2022) demonstrated that "Thymidine rescues ATR kinase inhibitor-induced deoxyuridine contamination in genomic DNA, cell death, and interferon- α /B" in cancer cells and CD8+ T cells. This manuscript also showed that ATR is expressed at the limit of detection in naïve CD8+ T cells and is massively induced in proliferating CD8+ T cells. The current manuscript repeats and significantly advances these findings and these published manuscripts should be cited. I don't understand why they weren't.

Reviewer #2 (Remarks to the Author): with expertise in cancer immunology

The manuscript by Hardaker et al., reports the findings that ATR inhibitor ceralasertib exhibits anti-tumor activity in a CTL-dependent manner and this anti-tumor activity is separate from its direct suppressive activity on tumor cells. Of interest is that ceralasertib inhibits proliferating CTLs. However, this inhibitory activity on CTL is rapidly reversed after the drug is taken off. Mechanistically, ceralasertib activates the IFN-I pathway, likely in myeloid cells in the tumor microenvironment, to exert its activity in combination with PD-1 blockade.

Ceralasertib and PD-1 blockade combined immunotherapy has demonstrated promising efficacy in melanoma and lung cancer patients who are refractory to immunotherapy. The accompanying manuscript clearly shows the efficacy of the combinational PD-1 blockade and ceralasertib in refractory cancer patients.

The efficacy studies in mouse tumor models in this manuscript are well designed and controlled and the efficacy is clearly demonstrated. The efficacy studies in mouse tumor models, although important, are not that significant considering what have been observed in human cancer patients. Elucidating the cellular and molecular mechanisms underlying celalaseritib anti-tumor activity is of significance. Therefore, the novelty and significance of this manuscript depend on its mechanistic findings.

Major comments:

The authors showed decreased proliferating CTLs, M-MDSCs, and TAMs in cerelasertib-treated tumor-bearing mice. The IHC data is clear. For the flow cytometry data, it is important to show the reference: the % cell is based on total cells or CD45+ cells.

Although the authors showed changes of cells in the proliferating cell population, but not in the resting cell population, and the decreased cells rapidly rebound after drug is off, there is still possibility that the T and myeloid cell progenitors are affected. It is therefore important to determine whether T cell progenitor cells in the bone marrow and thymus, and myeloid progenitor cells in the BM are affected by ceralasertib in tumor-bearing mice. This will determine the potential immune toxicity.

The authors determined that the myeloid cell IFN-I pathway is a major mechanism of ceralasertib action. Several recent reports have shown that PD-1+ myeloid cell IFN-I signaling pathway plays a key role in CTL tumor infiltration and recruitment. It appears that PD-1 expression is affected in the treated mice by ceralasertib. This pathway needs to be explored or at least needs to be analyzed/discussed.

The author stated that "This upregulation of ATR potentially renders T-cells more dependent on ATR to initiate repair of damaged DNA, which accumulates in cells that have undergone more cell divisions". What causes DNA damages in the replicating T cells? PMN-MDSCs were not changed and M-MDSC and TAM are decreased. Are there different DNA damage repair mechanisms or ATR expression profiles in these myeloid cell subsets? Are there different DNA damages in the proliferating T cells/M-MDSC/TAM as compared to PMN-MDSC?

Reviewer #3 (Remarks to the Author): with expertise in cancer immunology, computational biology

General comments

The manuscript by Hardaker and colleagues describes a study on the mechanisms of ATR inhibitor ceralasertib on potentiating the effects of checkpoint blockers. The authors show that ATR inhibition affects several cells in the TME including suppression of CD8T+ cell proliferation, enrichment for activated DCs and depletion of suppressive myeloid cells. While the used methods are sound and the presented data compelling there are two important issues that need to be addressed in order to provide a stronger support for the rationale of

combination therapy. First, in general for cancer therapies and specifically for cancer immunotherapies it became clear that the cellular context is of importance and that tumor agnostic approaches are risky. The authors used mostly two CRC mouse models, CT26 and MC38, whereas the human data is from HNSCC patients (and one clinical trial from NSCLC patients). Moreover, the CT26 and MC38 were used interchangeably (e.g. IFN γ effects with the MC38 model, whereas cyTOF, scRNA-seq and bulk RNAseq with the CT26 model), albeit they have different sensitivities to checkpoint blockade. Second, pleiotropic effects of the ATR inhibitor indicates that there is considerable rewiring of the cell-cell crosstalk in the TME. The authors used scRNA-seq data but did not carry out ligand-receptor interaction analysis which could pinpoint heterocellular crosstalk in the TME (which could shed light on potential mechanisms). This analysis should be carried out including validation for selected candidates.

Specific comments

1. Along the same line as above, all mouse experiments used transplanted models, which do not mimic the native TME. Data from experiments with genetic models could provide further evidence.
2. The authors used only anti PD-L1 antibody in the mouse studies whereas in the clinical trials both anti PD-L1 (for HNSCC) and anti PD1/PD-L1 for NSCLC were used. What is the impact of the combination therapy targeting different checkpoints molecules?
3. Figure 1a: it seems that for the MC38 model the ceralasertib is more efficient than the combination ceralasertib/antiPD-L1. Was this statistically significant? Experiments with nude mice could provide some hints.
4. Discussion, page 18, second paragraph: discussion regarding the clinical trials should be omitted, as it does not contribute to the story.

Reviewer #4 (Remarks to the Author): with expertise in cancer immunology

The manuscript by Hardaker E. and colleagues describes how the ceralasertib (cerala), an ATR inhibitor used in the clinics, enhances the efficacy of PD-L1 immunotherapy in different preclinical models of tumor. The authors first evaluated the impact of cerala, either alone or in combination with antiPD-L1, on tumor growth unveiling a significant increase in peripheral or infiltrating T cell frequency, proliferation and effector function. Then the authors moved on to evaluate the effect of cerala on CD8 and CD4 T cells, reporting a major impact of cerala on proliferating and memory T cells as compared to the naïve counterpart. This effect associated with a drop in the frequency of T cells during the treatment that was rapidly restored upon treatment termination. This effect was also present in T cells infiltrating the tumor as revealed by CyTOF and scRNAseq. Treatment with cerala also shaped the myeloid compartment of the TME, with a marked up-regulation of dendritic cell markers, notably MHCII, CD40 and CD80. Finally, the authors identify the IFN type I as the signaling pathway that contributed and explained cerala-associated effects both in preclinical models and in humans.

The manuscript is of relevance given the positive and encouraging results coming from clinical studies in which patients with either melanoma or lung cancer, who progressed in immunotherapy, were treated with a combination of cerala+antiPD-L1. Therefore, understanding cerala mode of action is relevant and the manuscript timely.

However, several aspects of the manuscript remain unclear and some of the conclusions are not supported by the data. Several experiments were performed in one tumor model, while others in a different model, making the manuscript difficult to follow. Here below some considerations, that might help to make the message clear.

Specific issues

1. As stated by the authors, and showed in Fig. 1A, cerala has a potent anti-tumor effect as a single agent. It seems that combination with antiPD-L1 is beneficial only for some tumors (e.g. A20

model in Fig. A). However, in the manuscript the authors focus on CT26 and MC38.

2. There is a missing link between the preclinical models used (mainly CT26 and MC38) and the human data obtained from HNSCC.

3. In Fig.1A, CT26 plot, the statistical analysis between α PD-L1 and the combo, and cerala vs vehicle, is missing; this makes difficult to conclude that cerala is active as single agent and the combo is more effective than monotherapy.

4. In Fig. 1C, does cerala have immune-independent effects? The statistical analysis between α CD8 and cerala+ α CD8 should be provided.

5. Fig.1E is difficult to interpret and the explanation in the text is difficult to understand.

6. In Fig. 1F, MC38 tumor CD8, it is clear that this tumor is not infiltrated by CD8 T cells, so conclusion derived from Fig. S7 on CD4 and CD8 infiltrating the MC38 are not easy to reconcile.

7. If we consider the grow curves (cerala only) presented in Fig. S2 and Fig.S3 and concerning CT26 and A20 models, they are very different. How could this be explained?

8. In Fig.S4C, the authors show a replica of experiments presented in Fig.S4A-B. However, the growth of the tumor seems completely different. Additionally the gray shadow in "cerala continuous" Fig.S4B is different than the corresponding presented in A and C. For how long did the authors monitor the growth of tumor in "cerala continuous"?

9. In Fig. 2C, the authors show the different impact that cerala has on CD8 naive, memory and CD4. The WB presented in Fig. 2E should be done on the separated populations not on total CD3.

10. In Fig. S6, the authors analyzed the effect of celara on PBMCs derived from HDs. While it is evident the effect on pKAP1, why there is no effect on pATR?

11. Should be informative to isolate CD8 and CD4 from patients treated with cerala to evaluate the pKAP1. Has this been done?

We thank our reviewers for their positive assessment of our work. We are also thankful for their thoughtful and constructive comments and criticism. In response, we performed additional experiments, provided clarifications in the text of the manuscript and fixed errors and omissions. Point by point responses to our reviewers' comments are provided below.

Reviewer #1

Hardaker et al. show that ATRi AZD6738 modulates the tumor microenvironment and potentiates responses to checkpoint blockade. The paper is a true tour de force and will be highly impactful, particularly if published with the manuscript describing results from the HUDSON trial in NSCLC (I am not a reviewer of this clinical paper). Two great strengths of the paper are the use of both preclinical mouse models and human specimens, and the analyses of the impact of Ceralasertib in different cell types. The ATR kinase inhibitors are a very promising class of drugs. But as this paper points out, schedule is absolutely critical.

I would argue that this manuscript sets a new standard for publication of responses to DNA damage response inhibitors in vivo. It has immediate clinical implications for a large number (10's) of ongoing clinical trials of ATR kinase inhibitors, both industry and NCI ETCN supported.

We thank our reviewer for this generous assessment. Clinical manuscript from the HUDSON trial is currently provisionally accepted for publication in Nature Medicine subject to minor clarification to the data and editorial revisions.

Minor weaknesses:

Figure 2e: I can't find the time point when activated CD8+ T cells were harvested. The ATM result is curious. I have not seen ATM phosphorylation and higher ATM expression in naïve CD8+ T cells than proliferating CD8+ T cells before. This result is different to that described by Sugitani et al. Cell Reports 2022. The authors should comment on this.

We have added the time point to the text and figure legend. There may not actually be such a significant difference in total ATM expression between non-stimulated and stimulated T cells as the protein loading (H3) is higher in unstimulated sample. While this reinforces the point of very low/no expression of ATR and DDR signaling in unstimulated cells it may exaggerate and overestimate the total ATM expression difference. However, we would also note that we have observed evidence elevated ATM expression in unstimulated naïve T-cells from four different human donors. Western blots shown below (**Figure for Review 1**) and in **Fig. S6**, with normalization to protein content and tRPA expression broadly equal across all samples.

Figure for Review 1. Western blot images from Supplementary Fig. S6 showing higher ATM expression in unstimulated compared to stimulated T-cells from 4 different human donors.

In reference to the Sugitani et al. data, the authors had loaded equivalent numbers of cells, but it is possible that there was higher overall protein loading in the proliferating vs. naïve T-cell lanes, when comparing Fig.1B and Fig.S1 ponceau stain (highlighted below), which may appear to reduce any difference in ATM expression (**Figure for Review 2**). Therefore, it may be the case the ATM expression may actually be broadly similar between studies than first appears.

We have edited the manuscript by removing the text suggesting a reduction in ATM and allow the readers themselves to assess the figures.

Figure for Review 2. Western blots comparing naïve and proliferating mouse T-cell from Sugitani et al. 2022. Left, DDR protein blots (Fig. 1B) and right, ponceau stain protein loading (Supplementary Fig. S1) which may suggest higher protein content loaded for proliferating T-cell lanes.

CT26/BALBc and MC38/C57BL6 are used to model T cell and myeloid responses to Ceralasertib. If the myeloid analyses were performed in both models, it would be good to show the CT26/BALBc data in the supplement. I think it's good to show that the impact of a pharmacologic on the immune system is different in different mouse models, as it almost certainly will be in humans.

In response, we added the analysis on myeloid cells in CT26 tumor-bearing mice. The results are now added to **Supplementary Figure 9** and as **Figure for Review 3**. The data are similar to those obtained in MC38 models (decrease in the presence of tumor associated macrophages and lack of the changes in PMN-MDSC).

Figure for Review 3. Impact of ceralasertib on myeloid cells in CT26 tumor-bearing mice. Mice were treated 8 days post implantation with 50 mg/kg q.d. of ceralasertib for 7 days. Tumor infiltrating macrophages (TAM), M-MDSC, PMN-MDSC, and DC were analysed by flow cytometry and calculated per gram of tissue. PMN-MDSC: CD11b⁺Ly6G^{hi}Ly6C^{lo/neg}, M-MDSC:CD11b⁺Ly6G⁺Ly6C^{hi}, TAM: F4/80⁺CD11b⁺Ly6G⁺Ly6C^{low}, DC:CD11c⁺MHC II⁺. *P<0.05, ** P<0.01 in one way ANOVA. Results of individual mice, mean and SD are shown. N=9.

Vendetti et al. JCI Insight, 8(4):e165615 (2023) demonstrated that the “Schedule of ATR inhibitor AZD6738 potentiates or abolishes anti-tumor CD8+ T cell responses to radiotherapy.” This manuscript also showed that intermittent Ceralasertib suppressed proliferation of CD8+ T cells on treatment which was rapidly reversed off treatment and that Ceralasertib potentiates the induction of IFN1 after radiation. Sugitani et al. Cell Reports, 20, 40(12):111371 (2022) demonstrated that “Thymidine rescues ATR kinase inhibitor-induced deoxyuridine contamination in genomic DNA, cell death, and interferon-α/β” in cancer cells and CD8+ T cells. This manuscript also showed that ATR is expressed at the limit of detection in naïve CD8+ T cells and is massively induced in proliferating CD8+ T cells. The current manuscript repeats and significantly advances these findings and these published manuscripts should be cited. I don’t understand why they weren’t.

We thank our reviewer for highlighting the omission of these two paper that were published during the submission/review process of our paper. We have added the references to the paper and highlighted them in the discussion with the following:

“During the submission of this manuscript two papers from independent research groups reported on the schedule dependency on ATR inhibitor treatment. Vendetti et al. (in the context of radiation therapy combination) showed that intermittent ceralasertib treatment suppressed proliferation of CD8+ T cells on treatment, which was rapidly reversed off treatment, and that ceralasertib potentiated the induction of IFN1 after radiation (Vendetti et al.). In addition, Sugitani et al. reported that proliferating mouse CD8+ T cells *ex vivo* demonstrated hallmarks of high replication stress and rely on ATR-dependent signaling for suppressing deoxyuridine (dU) contamination in genomic DNA, RNA-DNA polymerase collisions, IFN-α/β induction and viability (Sugitani et al.). These data are broadly consistent with our findings and demonstrated that the effects of ATR inhibition are applicable to other model systems or combinations.”

Sugitani, N., et al. Thymidine rescues ATR kinase inhibitor-induced deoxyuridine contamination in genomic DNA, cell death, and interferon-alpha/beta expression. Cell Rep 40, 111371 (2022).

Vendetti, F.P., et al. The schedule of ATR inhibitor AZD6738 can potentiate or abolish antitumor immune responses to radiotherapy. JCI Insight 8(2023).

Reviewer #2

Ceralasertib and PD-1 blockade combined immunotherapy has demonstrated promising efficacy in melanoma and lung cancer patients who are refractory to immunotherapy. The accompanying manuscript clearly shows the efficacy of the combinational PD-1 blockade and ceralasertib in refractory cancer patients. The efficacy studies in mouse tumor models in this manuscript are well designed and controlled and the efficacy is clearly demonstrated. The efficacy studies in mouse tumor models, although important, are not that significant considering what have been observed in human cancer patients. Elucidating the cellular and molecular mechanisms underlying celalasertib anti-tumor activity is of significance. Therefore, the novelty and significance of this manuscript depend on its mechanistic findings.

We appreciate this assessment of our work and in additional experiments expanded mechanistic findings of the study.

Major comments:

The authors showed decreased proliferating CTLs, M-MDSCs, and TAMs in cerelasertib-treated tumor-bearing mice. The IHC data is clear. For the flow cytometry data, it is important to show the reference: the % cell is based on total cells or CD45+ cells.

We have revised the figures to provide this information.

Although the authors showed changes of cells in the proliferating cell population, but not in the resting cell population, and the decreased cells rapidly rebound after drug is off, there is still possibility that the T and myeloid cell progenitors are affected. It is therefore important to determine whether T cell progenitor cells in the bone marrow and thymus, and myeloid progenitor cells in the BM are affected by cerelasertib in tumor-bearing mice. This will determine the potential immune toxicity.

We have performed additional experiments to address our reviewer questions. First, we assessed changes in thymic precursors during the treatment. The results are presented in **Supplementary Figure 8** and **Figure for Review 4**. Seven-day treatment of MC38 tumor-bearing mice with cerelasertib caused marked decrease in the presence of all populations T-cell precursors in thymus (**Fig. Rev. 4A**) as well as all populations of CD4/CD8 double negative (DN) cells (**Fig. Rev. 4B**). Seven days off the treatment, the populations of thymocytes started recovery, which was evident at later stages of DN populations (DN-2-4, **Fig. Rev. 4B**) and population of double positive thymocytes (**Fig. Rev.4A**). Single positive CD4 thymocytes also demonstrated significant recovery (**Fig. Rev. 4A**). These results are consistent with the data on cerelasertib effect on T cells in periphery.

A

B

Figure for Review 4. Effect of ceralasertib on thymic T cell precursors. MC38 tumor-bearing mice were treated for 7 days with 25 mg bid (7-days on) and then left untreated for 7 days (7 days on/ 7 days off). Thymocytes were evaluated as shown on graph. **A.** The populations of thymocytes. **B.** Analysis of double negative (DN) CD4⁺CD8⁻ precursors. The number of cells were calculated per thymus. Individual data, mean and SD are shown. P values were calculated in One-way ANOVA with correction for multiple comparisons. *P<0.05, ** P<0.01; ***P<0.001; ****P<0.0001. N=5.

We also assessed the effect of ceralasertib on myeloid cell progenitors in bone marrow. Results are included to **Supplementary Figure S10** and **Figure for Review 5.** Treatment with ATR inhibitor did not significantly affect the presence of myeloid progenitors, which was consistent with lack of the effect on the presence of largest population of myeloid cells in periphery (PMN) in our study.

The authors determined that the myeloid cell IFN-I pathway is a major mechanism of ceralasertib action. Several recent reports have shown that PD-1⁺ myeloid cell IFN-I signaling pathway plays a key role in CTL tumor infiltration and recruitment. It appears that PD-1 expression is affected in the treated mice by ceralasertib. This pathway needs to be explored or at least needs to be analyzed/discussed.

To address this comment/question, we performed several additional experiments. First, we evaluated the effect of IFN-I on the expression of PD1 on T cells *in vitro*. CD4⁺ and CD8⁺ T cell isolated from spleens

A

B

Figure for Review 5. Effect of ceralasertib treatment on bone marrow myeloid progenitors. MC38 tumor-bearing mice were treated and evaluated as described in Fig.Rev.2. **A.** Populations of early myeloid progenitors. **B.** Populations of late myeloid progenitors. Individual data, mean and SD are shown. P values were calculated in One-way ANOVA with correction for multiple comparisons. *P<0.05, ** P<0.01; N=5.

of naïve mice were treated with IFN- β in the presence of CD3/CD28 activation. IFN- β caused strong up-

regulation of PD1 expression on T cells (**Figure 5** and **Figure for Review 6**). Second, MC38 tumor-bearing IFNAR1 knockout mice were treated with ceralasertib for 7 days and expression of PD1 was

Figure for Review 6. Effect of IFN-β on the expression of PD1 on T cells. CD4⁺ and CD8⁺ T cells were isolated from spleens of naïve mice were treated for 72 hours with different concentration of IFN-β. T cells were stimulated with CD3/CD28. Expression of PD1 was measured by flow cytometry. P values were calculated in One-way ANOVA with correction for multiple comparisons. *P<0.05, ** P<0.01; ***P<0.001; ****P<0.0001. N=3. The same results were obtained after 48 hour of treatment.

assessed on spleen and tumors. In contrast to wild-type mice, ceralasertib failed to up-regulate PD1 expression in IFNAR1 deficient T cells (**Figure 5** and **Figure for Review 7**). These results strongly indicate that PD1 up-regulation in ceralasertib treated mice was mediated by IFN-I. We added more discussion to the text of the manuscript.

Figure for Review 7. Effect of ceralasertib on the expression of PD1 on T cells. Spleen and tumor CD4⁺ and CD8⁺ T cells were evaluated WT (N=10) and IFNAR1 KO (N=6) MC38 tumor-bearing mice treated for 7 days with

ceralasertib (25 mg BID). Expression of PD1 was measured by flow cytometry. P values were calculated in Student's t-test. *P<0.05, ** P<0.01; ***P<0.001.

The author stated that "This upregulation of ATR potentially renders T-cells more dependent on ATR to initiate repair of damaged DNA, which accumulates in cells that have undergone more cell divisions". What causes DNA damages in the replicating T cells? PMN-MDSCs were not changed and M-MDSC and TAM are decreased. Are there different DNA damage repair mechanisms or ATR expression profiles in these myeloid cell subsets? Are there different DNA damages in the proliferating T cells/M-MDSC/TAM as compared to PMN-MDSC?

Our reviewer raised an interesting question about possible differences in ATR expression and function in different cells. The presence of ATR and other components of the DDR signaling machinery in mouse T-cells is has recently been shown in other independent studies (Sugitani et al. Cell Reports 20, 40(12):111371; 2022), and confirmed in our study. To address this question for myeloid cells, we isolated different population of cells from spleens of tumor-bearing mice and assess different components of DDR. ATR and ATM as well as other components of DDR or cycle cycle-proliferation are highly expressed in T cells isolated from spleen. In monocytes and macrophages ATR, ATM, CHK1 and H2AX as well as CyclinE were clearly detectable (albeit at lesser degree than in T cells). However, PMN were negative for ATR expression and a number of other DDR and cell cycle-proliferation biomarkers.

To complement this analysis performed in mouse cells we have also looked at the same biomarkers in PMN and monocytes isolated from human peripheral blood. Human PMN cells also did not express ATR or cell cycle markers. However, taking isolated monocytes into culture, and then stimulating differentiation with different cytokines resulted in upregulation of ATR, cell cycle biomarker.

Collectively, these data reinforce the notion that actively replicating T- or myeloid cell populations upregulate the ATR dependent replication stress pathway, required for unperturbed cell proliferation and genome stability, and thus may be selectively susceptible to ATR inhibition. This may explain the in vivo observations for the lack of effect of ceralasertib on the presence of PMN, which are predominantly not replicating, in treated mice. The underlying reasons for these differences are not clear at this moment but will be focus on our future investigations. The results are included to the manuscript as **Supplementary Figure 10** and **Figure for Review 8**.

Figure for Review 8. Components of DDR in different cells. A. Spleen cells from MC38 tumor-bearing mice were sorted into indicated populations, lysed and immunoblotted for indicated DDR and cell cycle markers. Mouse CT-

26 cells were used to generate positive control lysates for DDR signaling using 2 mM HU. B. Human CD15+ PMN and CD14+ monocytes were isolated from healthy donors and either lysed immediately or cultured ex vivo for 6 to 8 days as indicated, before lysis and immunoblotting for the indicated DDR and cell cycle markers.

Reviewer #3

While the used methods are sound and the presented data compelling there are two important issues that need to be addressed in order to provide a stronger support for the rationale of combination therapy. First, in general for cancer therapies and specifically for cancer immunotherapies it became clear that the cellular context is of importance and that tumor agnostic approaches are risky. The authors used mostly two CRC mouse models, CT26 and MC38, whereas the human data is from HNSCC patients (and one clinical trial from NSCLC patients). Moreover, the CT26 and MC38 were used interchangeably (e.g. IFN1 effects with the MC38 model, whereas cyTOF, scRNA-seq and bulk RNAseq with the CT26 model), albeit they have different sensitivities to checkpoint blockade. Second, pleiotropic effects of the ATR inhibitor indicates that there is considerable rewiring of the cell-cell crosstalk in the TME. The authors used scRNA-seq data but did not carry out ligand-receptor interaction analysis which could pinpoint heterocellular crosstalk in the TME (which could shed light on potential mechanisms). This analysis should be carried out including validation for selected candidates.

We appreciate these comments and provide more data and information to address them. We agree with our reviewer that multiple tumor models would provide more confidence in the results. We would like to point out that we used not only CT26 and MC38 models, but also 4T1 and A20 tumor models with similar results. We did not however perform all biomarker experiments in all models. However to address this and increase confidence in the data presented we have provided more biomarker data derived via different methods used in both CT26 and MC38 models. These data demonstrate that at least in these models grown in different mouse backgrounds (CT26 Balb/c and MC38 Black6) the major on and off treatment biomarker changes mediated by ceralasertib are reproduced. The additional results are provided in response to reviewer 1 as well as below.

The reviewer's point regarding need for lung cancer relevant model is well taken. Therefore, we performed additional experiments in a G12C ; p53^{-/-} transgenic mouse model of lung cancer generated by CRISPR modification of Ras and p53 in a Cas9 inducible mouse (OdIN) (<https://www.nature.com/articles/s41467-020-18548-9>). These data recapitulated the results obtained in transplantable models, with ceralasertib treatment increasing survival. **The details of the experiments and the results are provided below in response to the specific comment.**

To address the question regarding ligand-receptor interaction, we performed additional experiments. WT and IFNAR1 KO MC38 tumor-bearing mice were treated with ceralasertib for 7 days and scRNAseq was performed. Ceralasertib treatment induced pronounced changes of the TME cell-cell interactions. These changes in interactions were increased within the myeloid compartment and decreased in T-cells. The magnitude of differences was markedly reduced in IFNAR1KO mice (**Figure for Review 9**). This analysis demonstrated overall changes in ligands/receptors interactions and strong effect of ceralasertib mediated by IFN1. However, due to multiple redundant interactions it will not allow for identification of specific interactions. Therefore, we decided not to include these results to current manuscript. This will be the focus on our next study.

Figure for Review 9. Ligand/receptors interactions in mice treated with ceralasertib. Differential interaction strength between Vehicle and Ceralasertib (7 days, 25 mg BID) treated MC38 tumor-bearing WT and IFNAR1KO mice as calculated by CellChat. The y-axis are quantifying the sending interactions and the x-axis the receiver. Red indicates increased strength under treatment, blue indicates increased strength in Vehicle. Color scale is cropped to the range -1×10^{-5} to 1×10^{-5} and is the same for both plots. Cell clusters <30 cells in any of the four conditions were excluded, as well as clusters of dividing cells.

Specific comments

1. Along the same line as above, all mouse experiments used transplanted models, which do not mimic the native TME. Data from experiments with genetic models could provide further evidence.

This comment of our reviewer is well taken. In response, we performed extensive experiments in a GEMM of lung cancer. This model was described in our previous study (Lundin et al. Nature Commun. 2020, PMID: 32994412). This model is based on doxocyclin inducible CAS9 mice. Tumors were induced by exposure of lungs to AAV containing guided RNA to provide for genome editing with deletion of *p53* gene and introduction of mutant *Kras* (*Kras*^{G12C}). Following viral induction mice developed lung tumor lesions by 6-7 weeks after the exposure. Mice were treated with ceralasertib (7 days on/7 days off) starting on day 50 and survival of mice was assessed. As shown in **Figure for review 10A** treatment significantly improved survival of these mice.

Analysis of tumors isolated from ceralasertib treated mice following a single- 7 days on/7 days off/1 day on cycles showed this was associated with significant decrease in alveolar macrophages and no change proliferating T cells (**Fig. Rev. 10B**), which was consistent with the data obtained in transplantable models. Analysis of IFN- γ signature in tumors demonstrated up-regulation of the signature (**Fig. Rev. 10C**). These results are included into manuscript as **Figure 6**.

2. The authors used only anti PD-L1 antibody in the mouse studies whereas in the clinical trials both anti PD-L1 (for HNSCC) and anti PD1/PD-L1 for NSCLC were used. What is the impact of the combination therapy targeting different checkpoints molecules?

Although it is interesting to compare the effect of combination of ceralasertib with different check-point inhibitors, this is very difficult to accomplish in a framework of a single study. We were focused on PD-L1 since this molecule was used in phase II clinical trials (HUDSON - accompanying clinical manuscript) and is currently being tested in phase III trial (NCT05450692).

Figure for Review 10. Cerasertib treatment of *Kras*^{G12C};*p53*^{-/-} LUAD GEMM. A. Kaplan-Meier survival analysis of lung-tumour bearing mice treated with vehicle (N=7) or cerasertib (cerala) (N=8) (Day 15; 7days on / 7days off + 1day on). Percentage of surviving animals plotted against days post-tumor induction. * P<0.05. **B.** Proportion of population of proliferating T cell and macrophages in lung tumors. Results of individual mice, Mean and SD are shown (N=7 for control and N=8 treatment). P values were calculated in Student's t-test. *P<0.05. **C.** Heatmap of Interferon type I signature gene expression, calculated from FPKM normalized counts expressed above biological viable levels, log-transformed and normalized for gene lengths. The data was z-transformed to be visualized in heatmap.

3. Figure 1a: it seems that for the MC38 model the cerasertib is more efficient than the combination cerasertib/antPD-L1. Was this statistically significant? Experiments with nude mice could provide some hints.

There was no difference between the effect of cerasertib alone and combination with PD-L1. We added this information to the Figure 1a. In response to our reviewer's suggestion, we performed additional experiments with nude MC38 tumor-bearing mice. The results are added to the manuscript as **Figure 1d** and is shown as **Figure for Review 11**. Nude mice were treated exactly as described in the manuscript for WT mice (**Fig. 1a**). In contrast to WT mice, cerasertib alone or in combination with PD-L1 antibody did not affect tumor growth (**Fig. Rev. 11**) further indicating T-cell dependent mechanism of antitumor activity mediated by cerasertib in the context of these syngeneic tumor models.

Figure for Review 11. Effect of treatment on tumor growth in MC38 tumor-bearing mice. Nude MC38 tumor-bearing mice were treated as indicated on graph. The protocol was the same as for WT mice (Fig. 1a). Tumor growth in individual mice (n=10) and tumor growth rate are shown. Statistics was calculated in one-way ANOVA with correction for multiple comparisons. ns- not significant.

4. Discussion, page 18, second paragraph: discussion regarding the clinical trials should be omitted, as it does not contribute to the story.

We revised discussion part of the manuscript to scale down clinical trials discussion.

Reviewer #4

The manuscript is of relevance given the positive and encouraging results coming from clinical studies in which patients with either melanoma or lung cancer, who progressed in immunotherapy, were treated with a combination of ceralasertib+ α PD-L1. Therefore, understanding ceralasertib mode of action is relevant and the manuscript timely. However, several aspects of the manuscript remain unclear and some of the conclusions are not supported by the data. Several experiments were performed in one tumor model, while others in a different model, making the manuscript difficult to follow. Here below some considerations, that might help to make the message clear.

We thank our reviewer to pointing on some deficiencies of the study. In response, we addressed them all either by additional experiments or by providing clarifications of the experiments.

1. As stated by the authors, and showed in Fig. 1A, ceralasertib has a potent anti-tumor effect as a single agent. It seem that combination with α PD-L1 is beneficial only for some tumors (e.g. A20 model in Fig. A). However, in the manuscript the authors focus on CT26 and MC38.

Different models had different sensitivity to both monotherapy ceralasertib and combination treatment. To study the effects of ceralasertib we selected a panel of tumor models with differentiated immune cell repertoires therefore the effects of each agent reflect differences in intrinsic sensitivity across these models.

For instance, in the CT26 model, single agent ceralasertib did not have a statistically significant effect on tumor growth, whereas in combination with PD-L1 antibody differences were significant ($p < 0.01$) (Figure 2A). However over a number of experiments the intermittent monotherapy treatment with ceralasertib

did induce significant anti-tumor activity, regressing tumors in a subset of animals in each study (Fig 2E). In the 4T1 model, while single agent activity was significant ($p < 0.05$), the combination resulted in more robust anti-tumor activity effect ($p < 0.0001$). In that model, combination demonstrated strong benefit for survival of mice, whereas single agent did not have significant impact. In the A20 model, the combination had a significantly stronger effect on tumor growth than single agent. Collectively these data demonstrates that the mechanism is relevant across models.

Having observed an impact of ceralasertib on tumor immune responses in pre-clinical models and in clinical trials, we were trying to understand how ATR inhibition may prime immune responses in patients. To explore the effect of ceralasertib in across the models we focused primarily on understanding the how ceralasertib modulates the TME to contribute to promote an anti-tumor immune response. The HNSCC data presented in the manuscript serve to illustrate the effect of ceralasertib treatment schedule on peripheral immune cells, and during the on treatment phase the reduction in Ki67 positive T-cells in the tumor.

We recognize the limitations of all syngeneic models for predicting efficacy in a specific disease type. Here we have used these models to probe the immune cell changes associated with the monotherapy and combination treatment in different models.

This was driven by the results of the observations in clinical trials demonstrating potent combination effect.

2. There is a missing link between the preclinical models used (mainly CT26 and MC38) and the human data obtained from HNSCC.

As stated above the data from HNSCC patients merely seeks to illustrate the general impact that ceralasertib has on peripheral immune cell populations in patients in the on and off treatment phases, and on the presence of Ki67+ T-cells in tumor in the on-treatment phase. They are not designed to illustrate a HNSCC specific effect. Mechanisms of ceralasertib effect on TME are not specific to HNSCC samples addition biomarker data are presented in the accompanying paper provided to reviewers to give full context to this data. The clinical trial data derived from the HUDSON clinical trial where ceralasertib and durvalumab give benefit for patients progressing following previous chemotherapy/checkpoint treatment is provided in accompanied manuscript (accepted in principal by Nature Medicine) and recent abstract (Iyer, S., et al. Abstract CT039: Immunomodulatory effects of ceralasertib in combination with durvalumab in patients with NSCLC and progression on anti-PD-(L)1 treatment (HUDSON, NCT03334617). *Cancer Research* 83, CT039-CT039 (2023)). These also show similar effects of ceralasertib in NSCLC. Recent abstract on the data in patients with HNSCC also consistent with our observation (Jones, G.N., et al. Abstract CT198: Immunomodulatory effects of the ATR inhibitor ceralasertib in a window of opportunity biomarker trial in patients with head and neck squamous cell carcinoma. *Cancer Research* 83, CT198-CT198 (2023)).

Thus, currently there is strong evidence that ceralasertib affect TME in cancer patients and provide positive support for check-point inhibitors combinations. Our models help to understand the possible mechanism of this effect and suggest the direction of future clinical studies.

3. In Fig.1A, CT26 plot, the statistical analysis between α PD-L1 and the combo, and cerala vs vehicle, is missing; this makes difficult to conclude that cerala is active as single agent and the combo is more effective than monotherapy.

We apologize for this lack of clarity. In most figures we showed statistical data only if they were significant to avoid cluttering the graphs. Therefore, in all other cases the differences were not highlighted the change were not significant. We have revised the text of figure legends to explain more clearly “not significant unless marked with *”.

4. In Fig. 1C, does cerala have immune-independent effects? The statistical analysis between α CD8 and cerala+ α CD8 should be provided.

We apologize for the omission. We added statistics to the figure. Also, **Fig. S3** shows athymic nude mouse CT26 efficacy data where activity of ceralasertib was seen suggesting little immune-independent effects in this model. We also added more extensive data on athymic MC38 tumor-bearing mice demonstrated similar lack of ceralasertib activity in these mice (**Fig. Rev.11**) and **Fig. S3**. In the syngeneic tumor models provided in this manuscript ceralasertib treatment delivers anti-tumor benefit through modifying the TME. However, ceralasertib can also impact tumor cell proliferation, indeed we show that in a panel of lung tumor cell lines that ceralasertib in combination with IFN γ results in increased inhibition of tumor cell growth or survival. Our mechanistic summary figure brings these different concepts together to show how ceralasertib can reduce tumor cell growth by targeting different cell types within the tumor.

5. Fig.1E is difficult to interpret and the explanation in the text is difficult to understand.

We apologies and have revised the text of manuscript to make it clear.

6. In Fig. 1F, MC38 tumor CD8, it is clear that this tumor is not infiltrated by CD8 T cells, so conclusion derived from Fig. S7 on CD4 and CD8 infiltrating the MC38 are not easy to reconcile.

We see quite substantial infiltration of T cells in MC38 tumors consistent with the literature data, and our own published studies (<https://jitc.bmj.com/content/6/1/158>; <https://jitc.biomedcentral.com/articles/10.1186/s40425-019-0794-7>) as well as the data presented in Fig. S7. Because of some outliers in that figure, the scale is different, which creates an impression of inconsistency.

7. If we consider the grow curves (cerala only) presented in Fig. S2 and Fig.S3 and concerning CT26 and A20 models, they are very different. How could this be explained?

The experiments were performed at different times in different animal facilities, this ensures that the effects seen were robust and not unique to a model run in an individual facility. We consider it a strength of our study that we could reproduce data in similar tumor modes established at different facilities.

The data highlighted by the reviewer represents variation in tumor growth curves that is commonly observed when repeating studies multiple times. For instance, 100% tumor rejection in response to ceralasertib in one group of experiments and 80% in another. However, they all showed the same phenomenon. In Figure 2E such an analysis of merged data from multiple studies is shown, where the impact of the intermittent dosing strategy on incidence of tumor regressions across multiple studies is shown.

8. In Fig.S4C, the authors show a replica of experiments presented in Fig.S4A-B. However, the growth of the tumor seems completely different. Additionally the gray shadow in “cerala continuous” Fig.S4B is different than the corresponding presented in A and C. For how long did the authors monitor the growth of tumor in “cerala continuous”?

These data represent independent biological replicate experiments and we acknowledge some differences in growth rates but between experiments the relative pattern or trend of activity was similar and does not change interpretation. With regard to dosing and monitoring periods we have added a vertical line and text to clearly state when tumor size monitoring stopped early (due to tumor progression), as in Fig. S4B and S4C, and the grey shaded area only refers to actual dosing for each study. The data are included to new version of manuscript and are shown below as **Figure for Review 12**.

Figure for Review 10. Intermittent 7d-on/7d-off schedule improves anti-tumor activity. Individual animal CT26 tumor volume growth curves from 3 independent replicate experiments ($n = 10$ per experiment). Ceralasertib daily dosing periods with intermittent 7 days-on/7days-off (x2) dosing compared to continuous daily dosing (23-28 days) are indicated by grey shaded areas. Measures stopped for all animals in continuous dosing groups at the point indicated by a dotted vertical line.

9. In Fig. 2C, the authors show the different impact that cerala has on CD8 naive, memory and CD4. The WB presented in Fig. 2E should be done on the separated populations not on total CD3.

We apologize for the confusion, and we have clarified the text to make sure it is explicit that we observed no significant differences on the impact between proliferating CD8 naïve, memory or CD4 with ceralasertib and have added the calculated ceralasertib 50% growth inhibitory concentrations (IC_{50}) values to **Fig. 2D** to show this similarity (**Figure for Review 13**). Although these different T-cell populations have different absolute rates of proliferation (as shown by differences in Proliferation Index with DMSO) the relative IC_{50} response to ceralasertib is very similar. Because all cell populations behaved similarly with respect to ceralasertib IC_{50} response we felt there was no need to separate populations for western blot analysis.

Figure for Review 13. Updated Figure 2D in the manuscript. Anti-proliferative activity of cerasertib vs vehicle (DMSO) control on *in vitro* stimulated/proliferating naïve CD8⁺ T-cells (nCD8⁺), memory CD8⁺ T-cells (mCD8⁺) and CD4⁺ T-cells (tCD4⁺).

10. In Fig. S6, the authors analyzed the effect of celara on PBMCs derived from HDs. While it is evident the effect on pKAP1, why there is no effect on pATR?

We appreciate the reviewers' comment and in context of the entire the ATR/DDR signaling dataset we think the pathway is modulated by cerasertib, as expected, but the downregulation of pATR S1989 in Western blot was small and difficult to clearly interpret. We would point out that the pATR S1989 antibodies are typically poor, with low signals (unless stimulated with replication stress inducing DNA damaging agent) and high background noise, and often not used to assess baseline expression. But despite this we have probed pATR S1989 marker for completeness and did observe small reductions relative to the DMSO (vehicle) control, particularly evident at higher 3 μM concentration (consistent with growth inhibitory impact).

We routinely measure the direct downstream effects of ATR by assessing phosphorylation of its target CHK1 (pS345) which is an acute marker of ATR kinase activation or inhibition. Reduction in pCHK1 S345 upon cerasertib treatment is consistent with ATR pathway modulation. To exemplify this a zoomed in blot for *ex vivo* activated T-cells from human Donor 1 in Figure S6 showing some reduction in pATR and pCHK1 with cerasertib.

In addition, in recent publication by Sugitani *et al.* (Cell Reports 20, 40(12):111371; 2022) the authors have profiled isolated and *ex vivo* activated mouse T cells for evidence of ATR signaling and used the pCHK1 S345 marker by western blot. The authors showed inhibition of pCHK1 by cerasertib at 5 μM (Sugitani *et al.* Fig. 4C and Fig. 6C) which independently corroborate our findings. In this regard we have referenced this paper and have added comment on the authors data in the discussion.

Example blot from Sugitani et al 2022 Fig. 4C showing reduction of pCHK1 S145 in ex vivo activated mouse T-cells treated with ceralasertib (ATRi). “Immunoblot of MCM4 in nuclease insoluble fraction in proliferating CD8+ T cells treated low rN or high rN for 2 h prior to 5 μ M ATRi for 1 h. Immunoblots of CHK1, CHK1 phosphoserine-345, and GAPDH are from the soluble fraction.” Nucleoside supplementation was added in some conditions: Low rN 250 nM A, C, 991 G, and T. High rN is 15 μ M A, C, G, and 6 μ M T.

11. Should be informative to isolate CD8 and CD4 from patients treated with cerala to evaluate the pKAP1. Has this been done?

Unfortunately, pKAP1 analysis of isolated CD8 and CD4 from ceralasertib treated patients has not been done and these data cannot be added at this time. However, we agree this would be interesting and in future clinical-translational studies this may be possible to assess. To acknowledge this question we have added the following sentence in the discussion that states it would be interesting to look at the regulation of pKAP in T-cells derived human samples.

“*ex vivo* stimulated/proliferating T- and myeloid cells (but not non-proliferating naïve cells) upregulated ATR expression and showed evidence of DNA damage induction (e.g. pKAP1, pRPA, γ H2AX) following ATR inhibition. Further translational analysis e.g. induction of pKAP1 in isolated T- and myeloid cells from human patient studies treated with ceralasertib would be warranted.”

REVIEWERS' COMMENTS

Reviewer #1 (Remarks to the Author):

The authors have written an exceptional rebuttal that addresses all concerns. I believe this manuscript documents very significant and important translational research.

Reviewer #2 (Remarks to the Author):

The authors have performed new experiments to address this reviewer's comments. The new experimental data and revised text addressed my comments.

Reviewer #3 (Remarks to the Author):

The authors addressed the issues raised in the original review satisfactorily. Specifically, additional experiments were carried out as well as analyses of the data as suggested, and thereby the manuscript was considerably improved.

The Figure for Review 9 (ligand/receptor interactions) should be included in the supplementary material and briefly commented in the manuscript.

Response to reviewers' comments.

We thank all our reviewers for their effort in evaluating our work.

Reviewer #1 (Remarks to the Author):

The authors have written an exceptional rebuttal that addresses all concerns. I believe this manuscript documents very significant and important translational research.

We thank our reviewer for this positive assessment.

Reviewer #2 (Remarks to the Author):

The authors have performed new experiments to address this reviewer's comments. The new experimental data and revised text addressed my comments.

We thank our reviewer for this positive assessment.

Reviewer #3 (Remarks to the Author):

The authors addressed the issues raised in the original review satisfactorily. Specifically, additional experiments were carried out as well as analyses of the data as suggested, and thereby the manuscript was considerably improved.

We thank our reviewer for this positive assessment.

The Figure for Review 9 (ligand/receptor interactions) should be included in the supplementary material and briefly commented in the manuscript.

We respectfully disagree with our reviewer. We believe the data are too preliminary and inconclusive to be discussed in the manuscript. Our reviewer previously correctly pointed out that ligand/receptor interaction need to be validated. It will require extensive effort and we are planning to do it in the future. However, currently, in the absence of validation, the message from these results maybe misleading.